# ASGO: Adaptive Structured Gradient Optimization

**Kang An**[1]*, **Yuxing Liu**[2]*, **Rui Pan**[2], **Yi Ren**[3], **Shiqian Ma**[1], **Donald Goldfarb**[4], **Tong Zhang**[2]

[1]Rice University [2]University of Illinois Urbana-Champaign
[3]Meta Platforms, Inc. [4]Columbia University

{kang.an,shiqian.ma}@rice.edu, {yuxing6,ruip4,tozhang}@illinois.edu, yiren94@meta.com, goldfarb@columbia.edu

## Abstract

Training deep neural networks is a structured optimization problem, because the parameters are naturally represented by matrices and tensors rather than by vectors. Under this structural representation, it has been widely observed that gradients are low-rank and Hessians are approximately block diagonal. These structured properties are crucial for designing efficient optimization algorithms, but are not utilized by many current popular optimizers like Adam. In this paper, we present a novel optimization algorithm ASGO that capitalizes on these properties by employing a preconditioner that is adaptively updated using structured gradients. By a fine-grained theoretical analysis, ASGO is proven to achieve superior convergence rates compared to existing structured gradient methods. Based on this convergence theory, we further demonstrate that ASGO can benefit from low-rank gradients and block diagonal Hessians. We also discuss practical modifications of ASGO and empirically verify ASGO's effectiveness on language model tasks. Code is available at https://github.com/infinity-stars/ASGO.

## 1 Introduction

Numerical optimization algorithms, especially those that can efficiently train large foundation models [Devlin et al., 2018, Brown et al., 2020, Touvron et al., 2023a,b, Ouyang et al., 2022], play an important role in the modern machine learning field. Among them, adaptive gradient methods like AdaGrad [Duchi et al., 2011] and Adam [Kingma and Ba, 2014] are popular choices, gaining huge success in training state-of-the-art models in many tasks. These algorithms typically apply a diagonal matrix preconditioner to the gradient $g_t$ to update the deep neural network (DNN) parameters $w_t$;

$$w_{t+1} = w_t - \eta_t \Lambda_t^{-1} g_t, \text{ where } w_t \in \mathbb{R}^d, g_t \in \mathbb{R}^d, \text{ and } \Lambda_t \in \mathbb{R}^{d \times d} \text{ is a diagonal matrix.}$$

This coordinate-wise step size design has been theoretically verified to be effective as it can exploit the sparsity of the gradient vectors $g_t$ [Duchi et al., 2011]. Also, when the Hessian is well-approximated by a diagonal matrix whose diagonal entries have very different scales, adaptive gradient methods have been proven to be beneficial [Liu et al., 2024, Jiang et al., 2024a, Xie et al., 2024]. While these results seem to be convincing, common DNNs do not necessarily have sparse gradients or Hessians that are well-approximated by ill-conditioned diagonal matrices. Instead, if we take the matrix structure of gradients in DNNs into account, it has been widely observed that these structured gradients are usually low-rank [Zhao et al., 2021, Yang et al., 2023, Cosson et al., 2023], and the Hessians are well-approximated by block diagonal matrices [Collobert, 2004, Zhang et al., 2024a,b]. Since adaptive gradient methods like Adam, treat the parameters as vectors and ignore the matrix structure of gradients, they are generally unable to exploit these structured properties. This leads us to ask the following question: *How can we effectively use the matrix structure of gradients to exploit their low-rank and block diagonal properties?*

---

*Equal Contribution. Ordering for the first two authors is determined by a coin flip.

39th Conference on Neural Information Processing Systems (NeurIPS 2025).

One possible answer is provided by Shampoo [Gupta et al., 2018]:

$$W_{t+1} = W_t - \eta_t L_t^{-\frac{1}{4}} G_t R_t^{-\frac{1}{4}}, \text{ where } W_t, G_t \in \mathbb{R}^{m \times n} \text{ and } L_t \in \mathbb{R}^{m \times m}, R_t \in \mathbb{R}^{n \times n} \text{ are full matrices.}$$

The main motivation for such a design is that if we apply vectorization to the update, the Shampoo preconditioner is a single matrix that is the Kronecker product of $L_t^{-\frac{1}{4}}$ and $R_t^{-\frac{1}{4}}$, which approximates the full-matrix preconditioner for AdaGrad [Duchi et al., 2011]. However, the theoretical convergence of Shampoo is worse than that for AdaGrad or even SGD when the dimension is large. Also, Shampoo needs more memory and much heavier computation than adaptive gradient methods because it requires two preconditioners, making it less suitable for training large-scale DNNs.

In this paper, we provide an answer to the aforementioned question by proposing ASGO (**A**daptive **S**tructured **G**radient **O**ptimization), which significantly improves the convergence guarantees of Shampoo while requiring less memory and computation. In light of the analysis that demonstrates the benefits of adaptive gradient methods [Duchi et al., 2011, Liu et al., 2024, Jiang et al., 2024a, Xie et al., 2020], we use appropriate assumptions to enable a more fine-grained convergence analysis, that yields superior convergence results and an explanation of how ASGO can benefit from the low-rank and block diagonal properties of a given problem. We also discuss the connection between ASGO and Muon [Jordan et al., 2024], a structured gradient method based upon the steepest descent algorithm with respect to the spectral norm, to conjecture a relation between ASGO and Muon analogous to the relation between AdaGrad and SignSGD [Bernstein et al., 2018, Kunstner et al., 2023]. Furthermore, we develop an efficient design for query-key attention parameters in transformer models and examine its empirical performance on pretraining transformer model tasks. Our main contributions are summarized as follows.

- We propose the structured gradient based algorithm ASGO, theoretically analyze its convergence, and show that it converges faster than full-matrix AdaGrad and Shampoo.

- We further demonstrate that ASGO can effectively exploit the low-rankness of gradients as well as the approximate block diagonal property of Hessians that is typically observed in training DNNS, indicating good theoretical properties under realistic settings.

- We develop a practical implementation of ASGO that is efficient while enhancing ASGO's effectiveness. We further empirically validate the effectiveness and efficiency of this implementation on language model tasks, demonstrating the algorithm's great potential in real applications.

## 2 Related Work

**Adaptive Gradient Methods.**   Adaptive gradient methods that use diagonal preconditioners to speedup convergence are extremely popular for solving many real-world optimization problems. To the best of our knowledge, the first method of this kind for machine learning, AdaGrad [Duchi et al., 2011, Streeter and McMahan, 2010], was developed based on rigorous theory that showed the benefits of using this kind of preconditioner. Adam [Kingma and Ba, 2014, Loshchilov and Hutter, 2017] modified AdaGrad and has become the default choice for training large foundation models. In theory, it has been proven that adaptive gradient methods can benefit from sparse gradients and approximately ill-conditioned Hessians [Duchi et al., 2011, Liu et al., 2024, Jiang et al., 2024a, Xie et al., 2024]. It is worth noting that the original AdaGrad paper [Duchi et al., 2011] also proposed a version of AdaGrad that uses a full-matrix preconditioner instead of the diagonal one, which is believed to perform even better. However, this full-matrix AdaGrad method suffers from large memory costs for storing the preconditioner, while having no better convergence guarantee compared to diagonal AdaGrad and SGD under the assumption of objective function convexity as described in Section 4 below.

**Optimization with Matrix Structure.**   In the standard optimization literature (i.e., excluding areas such as conic optimization), it is common to consider variables as vectors. However, recently, optimization methods for machine learning that consider variables as matrices have been rapidly gaining attention. Adafactor [Shazeer and Stern, 2018], LAMB [You et al., 2019], and Adam-mini [Zhang et al., 2024b] consider the matrix or layer structure to help reduce the memory cost of Adam and enable more efficient training. Shampoo [Gupta et al., 2018] and KFAC [Martens and Grosse, 2015] are two pioneering works that approximate, respectively, the full-matrix preconditioner of AdaGrad and the **true** Fisher matrix (FM), using Kronecker products of smaller matrices, making the memory cost more affordable.   Unfortunately, the analysis in [Gupta et al., 2018] shows that

the rate of convergence for Shampoo is no better than that for the full-matrix AdaGrad. We note that there is another method, TNT [Ren and Goldfarb, 2021] that is very closely related to Shampoo. TNT was developed as a natural gradient method, approximating the **true** FM by the covariance of block-wise sampling-based gradients assuming that they are Tensor-Normally distributed. The main difference between TNT and Shampoo, is that TNT uses true FMs and inverses of their Kronecker factors, whereas Shampoo uses empirical FMs and the -1/4 power of their factors. More discussion on related recent and concurrent work is presented in Appendix A.

## 3 Our ASGO Algorithm

### 3.1 Notation and problem setting

Throughout this paper, we use capital letters such as $W$ to represent matrices and $[W]_{i,j}$ to denote the $(i,j)$-th entry of $W$. For an arbitrary matrix $W \in \mathbb{R}^{m \times n}$, we denote

- $\|W\|_{\mathrm{op}}$ as the spectral norm of a matrix $W$, i.e., its largest singular value;
- $\|W\|_*$ as the trace norm of a matrix $W$, i.e., the summation of its singular values, which is well-known as the dual norm of the spectral norm;
- $\|W\|_{\mathrm{F}}$ as the Frobenius norm of $W$, which also equals $\mathrm{tr}\left(W^\top W\right)$, where $\mathrm{tr}\left(\cdot\right)$ is the trace;
- $\|W\|_L \triangleq \mathrm{tr}\left(W^\top L W\right)$, where $L \in \mathbb{R}^{m \times m}$ is a real symmetric positive definite matrix.

For symmetric square matrices, $A, B \in \mathbb{R}^{m \times m}$, $A \preceq B$ denotes that $B - A$ is positive semidefinite and $A \prec B$ denotes that $B - A$ is positive definite. $\succ$ and $\succeq$ are defined accordingly. We study the following stochastic optimization problem:

$$\min_{W \in \mathbb{R}^{m \times n}} f(W) \triangleq \mathbb{E}_\xi \left[ f(W, \xi) \right], \tag{1}$$

where we only have access to a stochastic gradient oracle $\nabla f(W; \xi)$ at $W$.

### 3.2 ASGO (Algorithm 1)

We propose ASGO (**A**daptive **S**tructured **G**radient **O**ptimization) in Algorithm 1, which is an algorithm with a single-side preconditioner. Compared to full-matrix AdaGrad, ASGO preserves and utilizes the matrix structure of $W_t$ and $G_t$, avoiding the huge memory cost for storing its preconditioner. ASGO's preconditioner consists of a single matrix compared to the two matrices used by Shampoo, leading to the following main update rule:

$$W_{t+1} = W_t - \eta_t V_t^{-\frac{1}{2}} G_t, \text{ where } W_t \in \mathbb{R}^{m \times n} \text{ and } V_t \in \mathbb{R}^{m \times m} \text{ is a full matrix.}$$

ASGO only needs to store one preconditioner matrix and compute its matrix square root and inverse on each iteration, versus two matrices for Shampoo. On the other hand, ASGO's preconditioner may not be as good an approximation to the full-matrix AdaGrad preconditioner or the empirical Fisher matrix [Gupta et al., 2018, Morwani et al., 2024] as Shampoo's. However, as we shall see in the following sections, this design can exploit the low-rankness of gradients and the block diagonal nature of Hessians to achieve better convergence rates.

## 4 Nonsmooth Theory

Although DNNs are generally nonconvex globally, convexity may apply locally in some regions. Thus, convex analysis can be helpful for understanding the behavior of DNN training algorithms.

**Assumption 1** (Convexity). *$f(\cdot)$ is convex and $W_*$ is one of its minimizers.*

**Theorem 1** (Nonsmooth convergence). *Under Assumption 1, for Algorithm 1 with $\eta_t \equiv \eta = D_{\mathrm{op}}$, it holds that*

$$\frac{1}{T} \sum_{t=0}^{T-1} \mathbb{E}[f(W_t)] - f(W_*) \leq \frac{1}{T} \mathbb{E}\left[ \left\| \left( \sum_{t=0}^{T-1} G_t G_t^\top \right)^{\frac{1}{2}} \right\|_* \right] \cdot D_{\mathrm{op}} + \frac{\epsilon D_{\mathrm{F}}^2}{D_{\mathrm{op}} T},$$

*where $D_{\mathrm{op}} \triangleq \max_{0 \leq t \leq T-1} \|W_t - W_*\|_{\mathrm{op}}$ and $D_{\mathrm{F}} \triangleq \max_{0 \leq t \leq T-1} \|W_t - W_*\|_{\mathrm{F}}$.*

---

**Algorithm 1** ASGO (**A**daptive **S**tructured **G**radient **O**ptimization)

---

1: **Input:** $W_0 \in \mathbb{R}^{m \times n}$, schedule $\{\eta_t\}$ batch size $M \in \mathbb{N}$, and the number of iterations $T$,
   $\epsilon > 0$, ($\epsilon$ should be small, similar to the $\epsilon$ for Adam or AdaGrad.)
2: Initialize $V_{-1} = 0 \in \mathbb{R}^{m \times m}$
3: **for** $t = 0$ **to** $T - 1$ **do**
4:     Sample mini-batch $\mathcal{B}_t$ with $|\mathcal{B}_t| \equiv M$ uniformly
5:     $G_t = \frac{1}{M} \sum_{\xi \in \mathcal{B}_t} \nabla_W f(W_t; \xi)$
6:     $V_t = V_{t-1} + G_t G_t^\top$
7:     $\Lambda_t = V_t^{\frac{1}{2}} + \epsilon I_m$
8:     $W_{t+1} = W_t - \eta_t \Lambda_t^{-1} G_t$
9: **end for**

---

**Corollary 2.** *If we also assume an upper bound for each stochastic sub-gradient such that* $\mathbb{E}\left[G_t G_t^\top\right] \preceq Q^2$, *where* $Q \in \mathbb{R}^{m \times m}$ *is a positive definite matrix, Theorem 1 also implies*

$$\frac{1}{T} \sum_{t=0}^{T-1} \mathbb{E}[f(W_t)] - f(W_*) \leq \mathcal{O}\left(\frac{\|Q\|_* D_{\mathrm{op}}}{\sqrt{T}} + \frac{\epsilon D_{\mathrm{F}}^2}{D_{\mathrm{op}} T}\right).$$

**Remark 1.** *Note that we treat $D_{\mathrm{op}}$ as a constant here, but it may increase as the iteration number $T$ increases. This can be addressed, for example, by invoking a projection onto a bounded convex set $\mathcal{W}$ with respect to the norm $\|\cdot\|_{\Lambda_t}$ in each iteration, as in AdaGrad [Duchi et al., 2011]. $D_{\mathrm{op}}$ would then be bounded by the spectral norm of $\mathcal{W}$. However, since this projection is rarely used in training DNNs, we follow Gupta et al. [2018] and omit it in Algorithm 1. We refer interested readers to Appendix H for a discussion of how one can incorporate the projection operation and obtain $D_{\mathrm{op}}$ and $D_{\mathrm{F}}$ independent of $T$. More importantly, this theoretical bound depends on the trace norm of gradients and the spectral norm of weights, showing that the algorithm can make use of the low-rank property of gradients.*

One can easily check that this convergence rate for convex nonsmooth problems is $\mathcal{O}(1/\sqrt{T})$, the same as SGD [Zinkevich, 2003] (see below) and AdaGrad [Duchi et al., 2011].

**Comparison with SGD.** The convergence rate for SGD under the assumptions of Corollary 2 is:

$$\text{SGD:} \quad \frac{1}{T} \sum_{t=0}^{T-1} \mathbb{E}[f(W_t)] - f(W_*) \leq \mathcal{O}\left(\frac{D_{\mathrm{F}} \|Q\|_{\mathrm{F}}}{\sqrt{T}}\right),$$

where $D_{\mathrm{F}}$ and $\|Q\|_{\mathrm{F}}$ are the Frobenius norm upper bounds for the weights and gradients, respectively. By comparing this bound with Corollary 2, we have

- $\|Q\|_{\mathrm{F}} \leq \|Q\|_* \leq \sqrt{r_G} \|Q\|_{\mathrm{F}}$, where $r_G$ is the rank of $Q$. Thus when the $G_t$ are low-rank, or have very imbalanced singular values, $\|Q\|_*$ can be close to $\|Q\|_{\mathrm{F}}$;

- $D_{\mathrm{F}}/\sqrt{r_D} \leq D_{\mathrm{op}} \leq D_{\mathrm{F}}$, where $r_D = \max \text{ rank of}(W_t - W_*)$. Thus, when the $W_t - W_*$ are relatively high-rank or have lots of singular values of a similar scale, $D_{\mathrm{op}}$ can be much smaller than $D_{\mathrm{F}}$.

Therefore, ASGO should work well when $G_t$ are low-rank and $W_t - W_*$ are relatively high-rank.

**Intuition about ASGO in practice.** We argue that $G_t$ is low-rank and $W_t - W_*$ is often relatively high-rank in many practical tasks, revealing the potential of ASGO in applications. As we have discussed in Section 2, gradients are commonly low-rank in DNNs, as verified by Zhao et al. [2021], Yang et al. [2023], Cosson et al. [2023]. Meanwhile, given the huge success of LoRA [Hu et al., 2022] in fine-tuning foundation models, which employs low-rank total updates of $W$, there may seem to be a conflict to assume that $W_t - W_*$ has a high rank. However, it has been observed that $W_0 - W_*$ should be relatively high-rank to obtain a better result, at least in pretraining and some complex fine-tuning tasks for large foundation models [Lialin et al., 2023, Jiang et al., 2024b, Huang et al., 2025]. Further exploration of the connection between the rank of weight updates and the performance of algorithms is an interesting topic for future research.

**Comparison with Full-Matrix AdaGrad and Shampoo.** Shampoo and the full-matrix AdaGrad achieve the following convergence rates under the same settings as Theorem 1.

$$\text{Full-Matrix AdaGrad:} \quad \mathcal{O}\left(D_{\mathrm{F}} \sum_{j=1}^{m} \sum_{i=1}^{n} \sqrt{\sum_{t=0}^{T-1} [G_t]_{i,j}^2}\right)$$

$$\text{Shampoo:} \quad \mathcal{O}\left(\sqrt{r} D_{\mathrm{F}} \cdot \mathrm{tr}\left(\left(\sum_{t=0}^{T-1} G_t G_t^\top\right)^{\frac{1}{4}}\right) \cdot \mathrm{tr}\left(\left(\sum_{t=0}^{T-1} G_t^\top G_t\right)^{\frac{1}{4}}\right)\right).$$

We can check that Theorem 1 indicates a convergence speed that is at least $D_{\mathrm{F}}/D_{\mathrm{op}}$ times faster than full-matrix AdaGrad and $\sqrt{r_G} D_{\mathrm{F}}/D_{\mathrm{op}}$ times faster than Shampoo; (see Appendix E for proofs). This also provides theoretical evidence that single-side preconditioning may be better at exploiting low-rankness of gradients, yielding a faster convergence speed compared to Shampoo-like preconditioning.

**Remark 2.** *Proofs of all of the results in this section are given in Appendix E. We keep the matrix structure of $W_t$ and $G_t$ throughout our analysis, in contrast to standard analyses of the convergence of AdaGrad and Shampoo, which are based on vectorizations of $W_t$ and $G_t$. This is important for proving that ASGO can exploit the structured properties. We note that for simplicity in Algorithm 1 above, the preconditioner operates on left-side of $G_t$. However, in a practical implementation of ASGO, using the right-sided preconditioner is preferable unless the number of columns of $G_t$ is much greater than the number of rows. See the discussion toward the end of Section 5 and in Section 7.1.2 and Algorithm 2 in Appendix B.*

## 5 Smooth Theory

It is also important to study the performance of Algorithm 1 in smooth settings, as many real training tasks have been widely observed to be smooth, at least locally. Also, only in smooth settings can we properly describe the importance of batch size.

**Assumption 2** (Smoothness). *$f$ is 1-smooth with respect to $\|\cdot\|_L$, where $L \in \mathbb{R}^{m \times m}$ is a symmetric positive definite matrix and for any $X \in \mathbb{R}^{m \times n}$,*

$$\|X\|_L^2 \triangleq \mathrm{tr}\left(X^\top L X\right).$$

If $X = [x_1, \ldots x_n]$, where each $x_i \in \mathbb{R}^m$, and we vectorize $X$, i.e., $\mathbf{x} \equiv \mathrm{vec}(X) = (x_1^T, \ldots, x_n^T)^T$, and form the block diagonal matrix $\mathbf{L} \equiv \mathrm{diag}[L, \ldots, L]$, we obtain that $\mathrm{tr}\left(X^\top L X\right) = \sum_{i=1}^{n} x_i^\top L x_i = \mathbf{x}^T \mathbf{L} \mathbf{x}$. This means that Assumption 2 is equivalent to the existence of a symmetric matrix $L \in \mathbb{R}^{m \times m}, L \succ 0$ such that for any $w \in \mathbb{R}^{mn}$, $-\mathbf{L} \preceq \nabla^2 f_v(\mathbf{w}) \preceq \mathbf{L}$, where $f_v(\mathbf{w}) = f(W)$, $W = [w_1, \ldots, w_n] \in \mathbb{R}^{m \times n}$ and $\mathbf{w} \equiv \mathrm{vec}(W) = [w_1^T, \ldots, w_n^T]^T \in \mathbb{R}^{mn}$. Hence, it is closely related to the block-wise diagonal structure of the Hessian as observed by many researchers with each block corresponding to a column of $W$; (see Figure 3 in Appendix A). Also note that Assumption 2 implies the standard smoothness assumption with respect to the Frobenius norm by $\|\nabla f_v(\mathbf{w})\|_{\mathrm{op}} \leq \|L\|_{\mathrm{op}}$. This block-wise diagonal smoothness is an extension of the standard smoothness, which is related to but different from the diagonal anisotropic smoothness employed for analyzing sign-based and adaptive gradient methods [Bernstein et al., 2018, Liu et al., 2024].

**Assumption 3** (Variance). *Let $N_t \triangleq \nabla f(W_t; \xi) - \nabla f(W_t) \in \mathbb{R}^{m \times n}$ be the stochastic gradient noise. We assume that $\mathbb{E}[N_t] = 0$ and there exists a symmetric positive definite matrix $V$ such that*

$$\mathbb{E}\left[N_t N_t^\top\right] \preceq V^2.$$

One can check that Assumption 3 implies the standard variance bound $\mathbb{E}[\|N_t\|_{\mathrm{F}}^2] \leq \|V\|_{\mathrm{F}}^2$. The assumption shares some similarity with the coordinate-wise variance bounds in Bernstein et al. [2018], Crawshaw et al. [2022], Liu et al. [2024], in the sense that it allows a more fine-grained analysis. This matrix-form variance upper bound may better describe the real case since it takes the structure of the noise into account, which is relevant to matrix rank and other structural properties.

**Theorem 3** (Smooth Convergence). *Under Assumptions 1, 2 and 3, for Algorithm 1 with $\eta_t \equiv \eta = D_{\mathrm{op}}$ and a batch size of $M$, it holds that*

$$\frac{1}{T} \sum_{t=0}^{T-1} \mathbb{E}[f(W_t)] - f(W_*) \leq \frac{4 D_{\mathrm{op}}^2 \|L\|_*}{T} + \frac{2\sqrt{2} D_{\mathrm{op}} \|V\|_*}{\sqrt{MT}} + \frac{2\epsilon D_{\mathrm{F}}^2}{D_{\mathrm{op}} T},$$

*where $D_{\mathrm{op}} \triangleq \max_{0 \leq t \leq T-1} \|W_t - W_*\|_{\mathrm{op}}$, and $D_{\mathrm{F}} \triangleq \max_{0 \leq t \leq T-1} \|W_t - W_*\|_{\mathrm{F}}$.*

As discussed in Section 4, $D_{\mathrm{op}}$ and $D_{\mathrm{F}}$ can be bounded if we add a projection step in the update. The convergence rate specified in Theorem 3 is similar to the rate in Corollary 2 if the batch size $M$ is small, when the $\mathcal{O}(1/\sqrt{MT})$ term dominates the rate, and thus shares the same properties as we discussed in Section 4. This means that ASGO can benefit if the stochastic gradient noise $V$ is generally low-rank. Furthermore, when the batch size $M$ is large so that the $\mathcal{O}(\|L\|_* / T)$ term contributes significantly to the bound, Theorem 3 implies more, which we now discuss.

**Comparison with SGD.** The convergence rate of SGD [Garrigos and Gower, 2023] is:

$$\text{SGD:} \quad \frac{1}{T} \sum_{t=0}^{T-1} \mathbb{E}[f(W_t)] - f(W_*) \leq \mathcal{O}\left( \frac{D_{\mathrm{F}}^2 \|L\|_{\mathrm{op}}}{T} + \frac{D_{\mathrm{F}} \|V\|_{\mathrm{F}}}{\sqrt{MT}} \right),$$

where $D_{\mathrm{F}}$ is defined in Theorem 3. Since the comparison between the $\mathcal{O}(1/\sqrt{MT})$ term is generally consistent with the discussion in Section 4, we further compare the $\mathcal{O}(1/T)$ term here to provide some more intuition. Specifically, we have

- $\|L\|_{\mathrm{op}} \leq \|L\|_* \leq r_L \|L\|_{\mathrm{op}}$, where $r_L$ is the rank of $L$. Thus when $L$ is low-rank, or has very imbalanced singular values, $\|L\|_*$ can be close to $\|L\|_{\mathrm{op}}$;

- $D_{\mathrm{F}}/\sqrt{r_D} \leq D_{\mathrm{op}} \leq D_{\mathrm{F}}$, where $r_D = $ max rank of $(W_t - W_*)$. Thus when $W_t - W_*$ are of relatively high-rank or have lots of singular values of a similar scale, $D_{\mathrm{op}}$ can be much smaller than $D_{\mathrm{F}}$.

Therefore, we can see that in general, ASGO should work well when the Hessian can be well approximated by a block diagonal matrix with a low-rank $L$ for each block and $W_t - W_*$ are of relatively high-rank. Note that Hessians have been found to be low-rank in DNNs, especially after some steps of training [Sagun et al., 2016, 2017, Wu et al., 2020].

**Block diagonal structure of DNNs.** It has been widely observed that the Hessian of MLPs and other DNNs, are approximately block-wise diagonal [Collobert, 2004, Zhang et al., 2024a,b, Bahamou et al., 2022], as shown in Figure 2 in Appendix A. This block-wise diagonal structure naturally arises from the structure of MLPs, where all the parameters associated with a particular neuron (i.e., all elements of a row of $W$) are more closely related and whose corresponding sub-block of the Hessian matrix is denser and has elements that are larger in absolute value than a set of parameters that do not share such an association. Intuitively, the elements of the rows of $W$ seem to be more likely to be closely related than those of the columns of $W$. Based on Theorem 3, our algorithm should perform well in this block Hessian setting, showing great potential in real applications.

**Intuition on single-side preconditioners.** Since ASGO only uses a single-side preconditioner, it is important to determine on which side they should be applied. As demonstrated in Figure 2, a block in the Hessian commonly corresponds to all input weights into a neuron in the next layer, i.e., as a row of the weight matrix $W$. Therefore, based on the block-wise smoothness convergence results, we should put the preconditioner $\Lambda_t$ on the right-hand side of $G_t$, i.e., computing $\Lambda_t = V_t^{1/2}$, where $V_t = V_{t-1} + G_t^\top G_t \in \mathbb{R}^{n \times n}$. In this way, we can better exploit the superior adaptability of block diagonal Hessians with correct block partitions (each row of $W$ as a block). One may also refer to Algorithm 2 for more details of our implementation. We also provide empirical results that show that right-sided outperforms left-sided preconditioning in Section 7.1.2.

**Remark 3.** *The proof of Theorem 3 is given in Appendix F. The analysis is similar in its general outline to the smooth analysis for AdaGrad [Levy et al., 2018, Liu et al., 2024], but it is more complex because of the involvement of matrix operations in ASGO.*

## 6 Further Discussions on ASGO

**Connection with Muon.** Muon [Jordan et al., 2024] can be interpreted as a standard steepest descent algorithm utilizing the spectral norm [Bernstein and Newhouse, 2024b] with momentum. If we ignore the incorporated momentum, Muon computes

$$W_{t+1} = \operatorname*{argmin}_{W \in \mathbb{R}^{m \times n}} \left\{ \langle G_t, W - W_t \rangle + \frac{1}{2\eta_t} \|W - W_t\|_{\mathrm{op}}^2 \right\}, \tag{2}$$

preserving the matrix structure of the problem and naturally exploiting the structural properties because of the involvement of spectral norm in the steepest descent framework. This aligns with ASGO's exploitation of the structured properties of the gradient. Moreover, if we ignore the momentum in both the gradient and the preconditioner, we can see that ASGO is equivalent to Muon.[2] Interestingly, this equivalence between ASGO and Muon in both theoretical basis and algorithm design is analogous to that between diagonal AdaGrad and SignSGD [Bernstein et al., 2018, Kunstner et al., 2023]. In this sense, we may interpret ASGO and Muon as AdaGrad and SignSGD for structured gradients, respectively. Note that Shampoo also admits such a relation with Muon in algorithmic design. However, as discussed in Section 4, Shampoo has a worse convergence rate and thus fails to benefit from the structured properties as does ASGO. To some extent, this means that ASGO may be a more appropriate approach than Shampoo as such an analog to diagonal AdaGrad.

From this intuition, we may also conjecture what the nonconvex convergence rate of ASGO is based on what it is for Muon. Hence, we prove in Appendix G that this rate for Muon is:

$$\min_{0 \le t \le T-1} \|\nabla f(W_t)\|_*^2 \le \mathcal{O}\left(\frac{\|L\|_* \left(f(W_0) - f^*\right)}{T}\right), \tag{3}$$

where we assume that $f^* \triangleq \inf f(W) > -\infty$. Since diagonal AdaGrad and SignSGD have the same convergence rate in nonconvex settings up to logarithmic factors [Bernstein et al., 2018, Sun et al., 2023, Liu et al., 2024], we expect that ASGO has a nonconvex convergence rate comparable to (3) obtained by Muon up to logarithmic factors. It is an interesting future topic to prove this conjecture and further explore the nonconvex behavior of ASGO theoretically. Also, we note that as an intuitively smoother version of Muon, ASGO has good convergence properties in nonsmooth settings as shown in Section 4, where Muon, like SignSGD [Bernstein et al., 2018], may fail to converge [Karimireddy et al., 2019].

**Remark 4.** *In smooth settings, nonconvex convergence analysis of Muon has been presented in Li and Hong [2025]. These convergence results are established under a more general setting, involving gradient noise and momentum following the analysis in Cutkosky and Mehta [2020]. However, if we only look at the deterministic case, their result is worse than (3) because of the explicit dependence on the dimension $n$. The key here is that the standard smoothness condition with respect to the Frobenius norm is not a good fit for analyzing structured gradient algorithms like Muon or ASGO. Using Assumption 2, we obtain better convergence results for Muon in (3).*

**Practical Implementations of ASGO.** We also provide Algorithm 2, a practical implementation of ASGO, together with a memory-efficient diagonal variant of ASGO, named DASGO in Algorithm 4. DASGO is basically a light-weight optimizer with the preconditioner $\Lambda_t$ of ASGO being diagonalized, which is efficient in both memory and computations. To make ASGO effective and efficient in practice, a critical aspect is the efficient computation of the preconditioner's inverse square root, $V_t^{-1/2}$. Standard methods such as Singular Value Decomposition (SVD) are prohibitively expensive for large-scale applications. To ensure our optimizer is practical, we adopt the Newton-Schultz (NS) iterative method as summarized in Algorithm 3, which substantially reduces the wall-clock time per step. The applicability of this approach stems from the deep connection between the matrix inverse square root and the matrix sign function. As established in Thm 5.2 [Higham, 2008b], computing $A^{-1/2}$ is equivalent to extracting a sub-block from the matrix sign function of an augmented matrix:

$$\text{msign}\left(\begin{bmatrix} 0 & A \\ I & 0 \end{bmatrix}\right) = \begin{bmatrix} 0 & A^{1/2} \\ A^{-1/2} & 0 \end{bmatrix}.$$

This equivalence implies that the computational complexity of our inverse square root step is on the same order as the matrix operation in the Muon optimizer. It is worth noting, however, that the NS iteration in ASGO is applied to this $2 \times 2$ block matrix, leading to a slightly heavier per-iteration cost compared to Muon. Finally, we highlight that our framework is modular. Recent advances [Amsel et al., 2025, Grishina et al., 2025] compute the matrix sign function more efficiently than classical Newton-Schultz, for instance, by optimizing the iteration's hyperparameters. This modularity allows such methods to be readily incorporated. In our empirical implementation, we validated this by experimenting with both the classical Newton-Schultz and PolarExpress [Amsel et al., 2025] methods. Please refer to Appendix B for more details of other modifications in our practical implementation.

---

[2]A proof of this equivalence can be found in Appendix G.

# 7 Empirical Results

We empirically evaluated the effectiveness of ASGO (Algorithm 2) and DASGO (Algorithm 4) on pretraining and finetuning tasks for Large Language models (LLMs). We compared our methods against the established optimizers, AdamW [Kingma and Ba, 2014, Loshchilov and Hutter, 2017], Shampoo [Gupta et al., 2018], and Muon [Jordan et al., 2024]. Several important implementational details should be noted for fair comparison. First, since Muon is designed to operate exclusively on matrix parameters, we followed Jordan et al. [2024] and applied AdamW update rules to all 1D parameters within the Muon optimizer to ensure that it can handle the complete model. All experiments were conducted using NVIDIA V100s SMX2 and NVIDIA GH200 GPUs. Specifically, for the larger-scale pretraining of GPT2, we utilized a configuration of four GH200 GPUs, while other experiments were performed on a single V100 GPU.

## 7.1 Pretraining GPT2

To further evaluate the efficacy of ASGO and DASGO, we extended our investigation to larger-scale pretraining tasks. We adopted the configuration of the GPT2 model as described by [Karpathy, 2022], comprised of 12 Transformer blocks, each with a hidden dimension of 768. All 1D parameters like bias and layer normaliztion and the embedding layer were trained by AdamW. There are 12 projection layers with dimensions $768 \times 2304$ in the Transformer blocks. For each of these layers alone, Shampoo needs to store two preconditioner matrices of size $768 \times 768$ and size $2304 \times 230$. In contrast, ASGO requires storing only one preconditioner matrix of size $768 \times 768$. This substantial difference highlights ASGO's advantages in memory requirements compared with Shampoo. We trained the model for 2400 steps using a batch size of 64, 4 H100 GPUs, a sequence length of 512, and 8 gradient accumulation steps. This setup corresponds to a total token budget of: $512 \times 64 \times 4 \times 8 \times 2400 \approx 2.5$ Billion tokens. This budget represents approximately 20 tokens per parameter as suggested by the Chinchilla scaling laws [Hoffmann et al., 2022]. We used both the PolarExpress (PE) and Newton-Schultz (NS) algorithms to compute the inverse square root of $V_t$ in ASGO. The parameters that we used in these algorithms can be found in Section C. To ensure a fair comparison, we carefully tuned the learning rates and $\beta_2$ values (where applicable) for all optimizers. (Tuning details are provided in Section C.) Furthermore, we employed a learning rate schedule consisting of 240 linear warm-up steps followed by a cosine decay, which is a common practice for training LLMs. Figure 1 and Table 1 present a comparison of the training and validation loss achieved by ASGO and DASGO against Shampoo, Muon and AdamW.

Table 1: Pretraining GPT2 Train loss and Validation Loss

| Optimizer | Training Loss | Validation Loss |
|---|---|---|
| AdamW | 3.442 | 3.455 |
| Muon | 3.332 | 3.342 |
| Shampoo | 3.405 | 3.417 |
| **DASGO (Ours)** | 3.497 | 3.509 |
| **ASGO-NewtonSchultz (Ours)** | 3.335 | 3.347 |
| **ASGO-PolarExpress (Ours)** | 3.332 | 3.342 |

We now present the performance evaluation of ASGO against several baselines on the GPT-2 pretraining task. The final training and validation losses after 2400 steps are summarized in Table 1, and the full training loss dynamics are depicted in Figure 1. Although the results obtained using ASGO-PE and ASGO-NS are similar, ASGO-PE performs slightly better than ASGO-NS. These results clearly demonstrate that ASGO achieves state-of-the-art performance, matching the strongest baseline while significantly outperforming AdamW and Shampoo. Focusing on the final validation loss in Table 1, ASGO-PolarExpress achieves a validation loss of 3.342. This result is on par with the Muon optimizer (3.342). More importantly, ASGO's performance is substantially better than both the standard AdamW (3.455) and Shampoo (3.417).

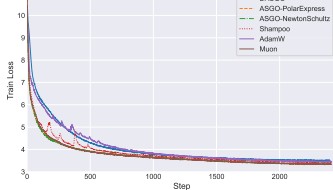

Figure 1: Pretraining GPT-2 Train Loss.

The training curves in Figure 1 provide deeper insights into the training dynamics. ASGO and Muon exhibit nearly identical training trajectories; both converge faster than AdamW. In contrast, while Shampoo displays loss spikes during the training process. This phenomenon suggests potential numerical instability even with preconditioner update frequency 1, a behavior not observed in ASGO.

These findings strongly support the efficacy of ASGO. It not only matches the final performance and rapid convergence of the highly-tuned Muon optimizer but also remedies the training instabilities observed in Shampoo.

However, DASGO (3.509) underperformed AdamW, Muon, Shampoo and ASGO in the GPT2 pretraining task. This performance gap was also observed in the pretraining of the smaller NanoGPT model. A primary reason for the difference between DASGO and ASGO likely stems from its diagonal preconditioning. By only retaining and using only the diagonal elements of the $GG^T$, DASGO essentially disregards the inter-dependencies between different parameter gradients from neurons within a layer, moving away from a true matrix-based adaptive approach towards a per-parameter scaling akin to vector-based methods. This highlights the value of preserving non-diagonal elements in the preconditioner matrices, particularly for capturing parameter interactions in attention-based architectures. Nevertheless, DASGO's significantly reduced memory footprint makes it a compelling option for resource-limited settings where computational efficiency is paramount.

### 7.1.1 Wall time comparison for pretraining GPT2

We now compare the Wall-Clock Times (WCTs) per training step for AdamW, Muon and ASGO, using the exact experimental setup from the GPT-2 pretraining task (Section 7.1).

Table 2: Wall-clock time per training step, difference compared with AdamW update.

| Optimizer | Single Train Step WCT | Time Difference |
|---|---|---|
| AdamW | 5.451 sec | 0 sec |
| Muon | 5.519 sec | 0.068 sec |
| **ASGO (Ours)** | 5.947 sec | 0.468 sec |

For this analysis, we used 10 iterations of the NS algorithm for ASGO instead of 5 iterations used for Muon's update, which are the settings required to achieve the optimal convergence performance reported previously. The measured WCT for a single training step (including forward, backward, gradient accumulation passes, and optimizer steps) is shown in Table 2. Although computing the inverse square root of $V_t$, in ASGO's optimizer step is more expensive than in AdamW, the forward and backward passes remain the dominant computational bottleneck. As shown in Table 2, a full training step with ASGO costs only 0.468 seconds more than the AdamW update. This overhead (8.6% increase over AdamW) is acceptable, and consequently, the total end-to-end training time is only marginally affected.

These tables show that although AdamW is computationally cheapest at the optimizer level, the total training times for all three methods are very similar. This demonstrates that the total training time may not be a main concern for ASGO. Note that ASGO achieves the smallest training loss while AdamW achieves the largest loss.

### 7.1.2 Ablation Study on the preconditioning side:

In Section 5, we theoretically motivated our choice of applying the single-sided preconditioner to the right side of the gradient (or its momentum), based on block-wise smoothness convergence results. To empirically validate this crucial design choice, we conducted an ablation study comparing the performance of ASGO with left-sided versus right-sided preconditioning. The experimental setup is identical to that used for the GPT-2 pretraining in Section 7.1. As presented in Table 3, the results demonstrate that right-sided preconditioning consistently and significantly outperforms its left-sided counterpart across all tested learning rates ($\{5e - 4, 1e - 3, 5e - 3, 1e - 2\}$).

This finding empirically confirms our theoretical intuition and highlights the clearly asymmetric impact of the preconditioning side. This observed asymmetry also suggests a potential limitation of Shampoo-like double-sided preconditioners. Beyond their higher memory and computational costs, their symmetric approximation (derived from a Kronecker product) may fail to capture the

true underlying structured curvature. This could lead to misplacing the preconditioning effect on the less critical side, resulting in less effective preconditioning. We hypothesize that this structural mis-approximation could be a contributing factor to the sub-optimal performance and training instabilities (e.g., the loss spike in Figure 1) observed in Shampoo.

Table 3: Performance comparison between precondition side

| Learning Rate | Validation Loss of left side | Validation Loss of right side |
|---|---|---|
| 5e-4 | 3.582 | 3.569 |
| 1e-3 | 3.468 | 3.447 |
| 5e-3 | 3.399 | 3.362 |
| 1e-2 | 3.412 | 3.386 |

## 7.2 Finetuning GPT2-Large on WikiText-2

To further assess the performance of our proposed optimizers, we conducted fine-tuning experiments on the GPT2-Large(774M) model [Radford et al., 2019] using the WikiText-2 dataset. We explored two distinct fine-tuning objectives:

- First, we fine-tuned GPT2-Large using the standard Causal Language Modeling (CLM) loss, which is the conventional approach for autoregressive language models.

- Second, we also fine-tuned the same GPT2-Large model on WikiText-2 but employed the Fill-in-the-Middle (FIM) training objective, following the setting in [Bavarian et al., 2022]. The FIM objective modifies the training process to enable the model to learn to infill text by rearranging document spans.

To account for statistical variability, we conducted five experimental runs with different random seeds under the same hyperparameter settings. Table 4 presents the average perplexity results after fine-tuning for 2 epochs. For both objectives, we employed a cosine decay learning rate scheduler. Hyperparameter search details and 95% confidence intervals are provided in Table 8 in Appendix C3.

Under the CLM objective, ASGO (13.88 perplexity) and DASGO (13.84 perplexity) achieved lower perplexity than both Muon (13.91 perplexity) and AdamW (14.01 perplexity). This suggests that both ASGO and its memory-efficient variant, DASGO, can be ef-

Table 4: Finetuning GPT2-Large Perplexity

| | AdamW | Muon | ASGO | DASGO |
|---|---|---|---|---|
| CLM | 14.01 | 13.91 | 13.88 | 13.84 |
| FIM | 17.46 | 15.93 | 15.66 | 15.45 |

fective for traditional LLM fine-tuning. When the model was fine-tuned using the FIM objective, we observed a similar trend, with ASGO (15.66 perplexity) and DASGO (15.45 perplexity) outperforming both Muon (15.93 perplexity) and AdamW (17.46 perplexity). Across both fine-tuning scenarios, ASGO and DASGO demonstrated competitive or superior performance compared to these baseline optimizers.

## 8 Conclusions

In this paper, we proposed a novel algorithm ASGO, which achieves significantly better convergence rates compared to full-matrix AdaGrad and Shampoo. Based on the theory, we demonstrated that ASGO can benefit from low-rank gradients and block-wise diagonal Hessians, which are widely observed structural properties of DNNs. We further proposed some practical modifications to ASGO, and verified its effectiveness empirically. Currently, ASGO still has two major limitations: 1) computationally intensive compared to Adam because of the matrix operation; 2) it is not straightforward to extend the algorithm to apply to tensors. We plan to look into these issues in future work.

## Acknowledgment

Shiqian Ma was partially supported by ONR grant N00014-24-1-2705, NSF grants CCF-2311275 and ECCS-2326591. This work was also partially supported by NSF grant No. 2416897 and ONR grant No. N000142512318, and used both the DeltaAI advanced computing and data resource, supported by NSF(award OAC 2320345) and the State of Illinois, and the Delta advanced computing and data resource, supported by NSF (award OAC 2005572) and the State of Illinois.

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

# A    Additional Related Work

**Optimization with Matrix Structure.**    Much recent work has focused on improving full-matrix AdaGrad and Shampoo. Feinberg et al. [2023] uses a sketching-based approach to approximate the full-matrix AdaGrad preconditioner with lower memory cost. Morwani et al. [2024] provides theoretical intuition and empirical evidence to claim that Shampoo should use the $-1/2$ power in its preconditioners to better approximate the Empirical Fisher (EF) matrix. Vyas et al. [2024] demonstrates that Shampoo is like doing Adafactor in the eigenspace of gradients and proposes a novel algorithm, SOAP, that performs Adam in this eigenspace. SOAP is observed to achieve better performance than Adam and Shampoo, but suffers from a high computation load per iteration. Galore [Zhao et al., 2024] and Apollo [Zhu et al., 2025] share a similar algorithmic design with SOAP, with extra focus on lowering memory costs. Muon [Jordan et al., 2024] follows this line of work, derived from the steepest descent (in the spectral norm) framework, which it shows to be scalable and effective in training large foundation models [Liu et al., 2025a]. This spectral descent method has its origin in restricted Boltzmann machines and discrete graphical models [Carlson et al., 2015a,b]. Large et al. [2024], Bernstein and Newhouse [2024a] propose the modular approach that has also been applied to improve Muon. Following Muon, Pethick et al. [2025] proposes the linear minimization oracle (LMO) framework and derives Scion based on it, showing good convergence hyperparameter transfer properties. More recently, Nguyen et al. [2025] proposes AdaDiag, which may be viewed as SOAP doing SVD without gradient accumulation. Liu et al. [2025b] proposes COSMOS, a combination of SOAP and Muon, that trades off between performance and computational efficiency.

**Rank of Gradients and Weight Updates.**    It has been widely observed that gradients are naturally low-rank in DNNs, even when a large batch size is employed [Gur-Ari et al., 2018, Zhao et al., 2021, Yang et al., 2023, Cosson et al., 2023]. This property has been widely utilized for computation and memory efficiency in training [Wang et al., 2018, Cosson et al., 2023, Zhao et al., 2024]. On the other hand, the rank of the total weight update $\Delta W = W_T - W_0$, depends a lot on the training method. If we use LoRA [Hu et al., 2022], $\Delta W$ is determined to be low-rank. However, in pretraining or even many complex fine-tuning tasks, LoRA's performance is much worse than methods that produce high-rank weight updates like full-parameter training, which is conjectured to be due to the weight update rank [Lialin et al., 2023, Jiang et al., 2024b, Huang et al., 2025].

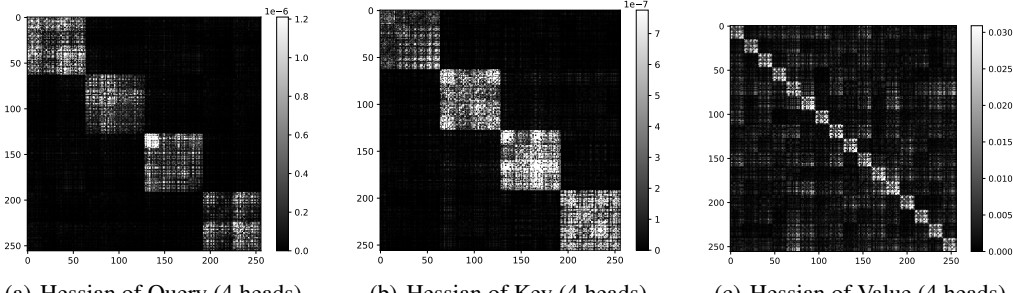

(a) Hessian of Query (4 heads)          (b) Hessian of Key (4 heads)          (c) Hessian of Value (4 heads)

Figure 2: This figure is from Zhang et al. [2024b]. It depicts the Hessian of different parameter blocks in a small Transformer at the 1% training step. The near-block-diagonal structure maintains throughout training. But different parameter blocks have different numbers of small dense matrices, where Query and Key correspond to the number of heads.

**Block-wise Diagonal Hessian.**    It has been observed that the Hessian of a neural network tends to be block-wise diagonal with each block corresponding to a neuron both in experiments and theory for small MLPs [Collobert, 2004]. Zhang et al. [2024a,b] recently numerically verified this property in small transformers, and as illustrated in Figure 3, further empirically showed that transformers may exhibit heterogeneity between blocks, while CNNs may not.

**Concurrent Work.**    When we were finishing writing this paper, we noticed that the paper [Xie et al., 2025], which had just appeared on arXiv, proposes and studies an algorithm, referred to as One-Sided Shampoo, that is identical to ASGO. Although the theoretical techniques and convergence results proved in [Xie et al., 2025] and here are very similar in general, there are still some notable differences between the two works. First, the motivations are different. In Xie et al. [2025], the

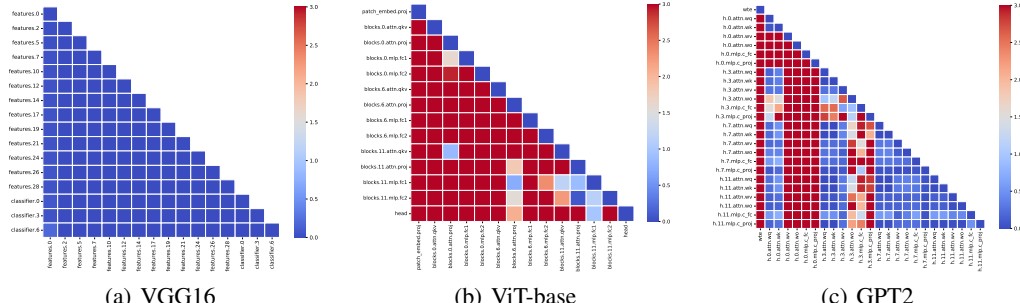

|  |  |  |
|---|---|---|
| (a) VGG16 | (b) ViT-base | (c) GPT2 |

Figure 3: This figure is from Zhang et al. [2024a,b], calculating the Jensen-Shannon (JS) distance between two eigenvalue densities of all possible block pairs at initialization. It shows that JS distance of blockwise spectra in CNNs is significantly smaller than that in Transformers.

authors develop their method from the unified preconditioning method framework AdaptReg [Gupta et al., 2017] and mainly highlight its superior convergence results compared to Shampoo and full-matrix AdaGrad. In our paper, we focus more on theoretically discussing how ASGO can utilize the structured properties of optimization problems including low-rank gradients and block-wise diagonal Hessians, to highlight the potential of ASGO as a practical algorithm for training deep learning models. Second, our empirical results provide evidence that ASGO can perform well in practical tasks, whereas Xie et al. [2025] focuses on convex settings and examines One-Sided Shampoo only on linear regression. Moreover, our implementation includes specialized designs for Transformer architectures (particularly query/key) to improve performance on deep learning training tasks. As a direct byproduct of our main algorithm, we also developed a lightweight diagonal version named DASGO to trade off memory consumption and performance. Third, Xie et al. [2025] focuses primarily on developing a general proof framework applicable to multiple optimizers including One-Sided Shampoo, whereas our work specifically examines the theoretical and practical benefits of ASGO. To conclude, both works contribute to a better understanding of ASGO/One-Sided Shampoo.

## B  More Discussions on ASGO

**A practical implementation of ASGO.**  We present a practical implementation of ASGO in Algorithm 2, with the input $W$ as a module in the network. Our implementation incorporates several common modifications to enhance computational efficiency and stability.

---

**Algorithm 2** A Practical Implementation of ASGO (for a layer $W^\ell \in \mathbb{R}^{m \times n}$ such that $m \geq n$)

---

1: **Input:** $W_0 \in \mathbb{R}^{m \times n}$, lr schedule $\{\eta_t\}$, momentums $\beta_1, \beta_2 \in [0, 1)$, batch size $M \in \mathbb{N}$, $\epsilon \in \mathbb{R}$ ($\epsilon$ should be small, similar to the $\epsilon$ for Adam.)
2: Initialize $M_{-1} = 0 \in \mathbb{R}^{m \times n}$, $V_{-1} = 0 \in \mathbb{R}^{n \times n}$     ▷ If $m < n$, then we set $V_{-1} = 0 \in \mathbb{R}^{m \times m}$
3: **for** $t = 0$ **to** $T - 1$ **do**
4:     Sample mini-batch $\mathcal{B}_t$ with $|\mathcal{B}_t| \equiv M$ uniformly
5:     $G_t = \frac{1}{M} \sum_{\xi \in \mathcal{B}_t} \nabla_W f(W_t; \xi)$
6:     $M_t = \beta_1 M_{t-1} + (1 - \beta_1) G_t$
7:     $V_t = \beta_2 V_{t-1} + (1 - \beta_2) G_t^\top G_t$     ▷ If $m < n$, $V_t = \beta_2 V_{t-1} + (1 - \beta_2) G_t G_t^\top$
8:     $\Lambda_t^{\text{inv}} = \texttt{SqrtInverseNewtonSchulz}(V_t)$     ▷ Run Alg. 3 to compute $V_t$ square root inverse
9:     $W_{t+1} = W_t - \eta_t \frac{0.2\sqrt{mn} M_t \Lambda_t^{\text{inv}}}{\|M_t \Lambda_t^{\text{inv}}\|_{\text{F}}}$     ▷ If $m < n$, $W_{t+1} = W_t - \eta_t \frac{0.2\sqrt{mn} \Lambda_t^{\text{inv}} M_t}{\|\Lambda_t^{\text{inv}} M_t\|_{\text{F}}}$
10: **end for**

---

(i) Exponential moving averages are used to update the preconditioner and incorporate momentum, following established practice, which results in Adam's improvements across various tasks over AdaGrad.

(ii) As demonstrated in Section 6, we use an efficient numerical algorithm (Algorithm 3) to compute the matrix square root for ASGO, which is mathematically equivalent to applying the Newton-Schulz iteration to the $2 \times 2$ block matrix $\begin{bmatrix} 0 & A \\ I & 0 \end{bmatrix}$. The parameters $a, b, c$ can be chosen as established

---

**Algorithm 3** SqrtInverseNewtonSchulz

---
1: **Input:** $X \in \mathbb{R}^{n \times n}$, parameters $a, b, c$ (can be chosen in ways like Jordan et al. [2024], Amsel et al. [2025], Grishina et al. [2025]), step number $K$, $\epsilon$
2: Obtain normalized input $Y \leftarrow \frac{X}{\alpha_X}$, with $\alpha_X = \|X\|_{\mathrm{F}} + \epsilon$
3: Initialize identity matrix $Z \leftarrow I_n \in \mathbb{R}^{n \times n}$
4: **for** $k = 1$ **to** $K$ **do**
5:     $A \leftarrow ZY$
6:     $B \leftarrow bA + cA^2$
7:     $Y \leftarrow aY + YB$
8:     $Z \leftarrow aZ + BZ$
9: **end for**
10: **return** $\frac{Z}{\sqrt{\alpha_X}}$     ▷ Return the approximation of $X^{-\frac{1}{2}}$, and $\sqrt{\alpha_X}\,Y$ is an approximation of $X^{\frac{1}{2}}$

---

in Jordan et al. [2024], Amsel et al. [2025], Grishina et al. [2025]. In each of the Newton-Schulz iteration, we perform 4 matrix multiplications, while the Muon Newton-Schulz iteration requires 3.

(iii) We adaptively select the preconditioning side. Instead of fixing the preconditioner to one side, we dynamically choose to precondition on the side corresponding to the smaller dimension of the gradient matrix $G_t$ in $\mathbb{R}^{m \times n}$. Specifically, if $m < n$, the preconditioner is formed from $G_t G_t^T$ and applied from the left; otherwise (if $m > n$), it is formed from $G_t^T G_t$ and applied from the right. Based on the intuition interpreted in Section 5, we also apply the preconditioner from the right when $m = n$.

(iv) We incorporate a similar update alignment to ASGO, as presented in line 9 of Algorithm 2. Specifically, we normalize the update of ASGO for each matrix weight $W$ by the root mean square norm (RMS norm) and multiplies 0.2 to match the RMS norm of Adam update, as suggested by Liu et al. [2025a]. In this way, the training becomes more stable and easier to tune.

(v) There are more possibilities in doing practical implementation for ASGO. For example, an alternative approach to improve the computational efficiency is doing matrix operations in a lazy way, i.e., the operations only happen every several iterations, which is widely employed in previous algorithms [Shi et al., 2023, Vyas et al., 2024, Zhao et al., 2024, Chen et al., 2024]. In this way, we can further lower the computational cost of ASGO, but more efforts on preventing performance degradation will be needed. We may also consider incorporating the variance reduction techniques as in Yuan et al. [2024], Cutkosky and Mehta [2020].

**Special Design for Transformers.** Furthermore, we introduce a specialized adaptation of ASGO for query and key matrices in attention layers. Recent work by Zhang et al. [2024b] has demonstrated that the Hessian structure of query and key layers differs significantly from conventional MLP layers. As illustrated in Figure 2, the number of dense blocks corresponds to the number of attention heads rather than output neurons. This observation aligns with the forward computation of multi-head attention, where attention scores are computed independently across different subspaces, suggesting that query and key parameters could be optimized in a head-wise manner. To leverage this insight, we reshape query and key parameters from matrices in $R^{n \times hd}$ to three-dimensional tensors $R^{h \times n \times d}$, and apply our optimization algorithm independently to each head's subspace. This restructuring reduces both memory consumption and computational complexity by decreasing the matrix size from $O(h^2 d^2)$ into $O(hd^2)$. For instance, in NanoGPT [Karpathy, 2022], this adjustment reduces the preconditioning size of a single query/key parameter from approximately $10^6$ elements into $10^5$. The empirical performance of this modification is evaluated in NanoGPT pretrain task in Section C.1.

**A Diagonal Variant of ASGO.** Drawing inspiration from Duchi et al. [2011], we also implement a variant of ASGO using diagonal matrices, which we denote as DASGO, presented in Algorithm 4. DASGO can be viewed as a lightweight version of ASGO, which eliminates the need to compute the inverse square root of full matrices, and reduces memory requirements to a level comparable with Adam-mini [Zhang et al., 2024b]. It also makes the choice of which side of $M_t$ to precondition unimportant in terms of computational effort, in contrast to ASGO. For DASGO, we choose to apply the diagonal preconditioner on the right side, aligning with the neuron architecture in DNNs. However, since DASGO only employs a diagonal preconditioner, it fails to recover the superior

theoretical properties of ASGO under block-wise diagonal Hessian settings . We further empirically examine this tradeoff in Section 7.

---

**Algorithm 4** Implementation of DASGO (**D**iagonal **A**daptive **S**tructured **G**radient **O**ptimization)

---

1: **Input:** $W_0 \in \mathbb{R}^{m \times n}$, $\epsilon \in \mathbb{R}$, lr schedule $\{\eta_t\}$, momentums $\beta_1, \beta_2 \in [0, 1)$, and batch size $M \in \mathbb{N}$
2: Initialize $M_{-1} = 0 \in \mathbb{R}^{m \times n}$, $v_{-1} = 0 \in \mathbb{R}^n$
3: **for** $t = 0$ **to** $T - 1$ **do**
4:     Sample mini-batch $\mathcal{B}_t$ with $|\mathcal{B}_t| \equiv M$ uniformly
5:     $G_t = \frac{1}{M} \sum_{\xi \in \mathcal{B}_t} \nabla_W f(W_t; \xi)$
6:     $M_t = \beta_1 M_{t-1} + (1 - \beta_1) G_t$
7:     $v_t = \beta_2 v_{t-1} + (1 - \beta_2) \operatorname{diag}(G_t^\top G_t)$
8:     $W_{t+1} = W_t - \eta_t M_t \operatorname{diag}(v_t + \epsilon)^{-\frac{1}{2}}$
9: **end for**

---

# C    Details of Empirical Experiments

## C.1    Pretraining NanoGPT

As an initial experiment, we compared the performance of ASGO, DASGO, Muon, Shampoo, and AdamW for training NanoGPT, which consists of 6 Transformer layers, 6 attention heads, and an embedding dimension of 384, on the Shakespeare character-level dataset with a sequence length of 256 tokens. For all algorithms, we trained the model for 20 epochs, where each epoch contained 128 steps, using a batch size 128 and the OneCycle learning rate (lr) schedule. In contrast to our experiments on GPT2, where we trained 1D and Embedding layer parameters as done in AdamW, we trained all the parameters independently in ASGO.

Before discussing hyperparameter (HP) tuning, we note that recent research [Shi et al., 2023, Morwani et al., 2024, Lin et al., 2024] has suggested treating the inverse order (IO) in Shampoo's preconditioner as a tunable HP rather than using the standard value of $-1/4$. Notably, Morwani et al. [2024] demonstrated that Shampoo with IO$= -1/2$ provides a superior approximation of the full-matrix AdaGrad optimizer. Consequently, we treated the IO as a Shampoo HP in our pretraining experiments. Additionally, Shampoo's performance is highly dependent on the initialization of its preconditioner matrices, as this is crucial for accurately approximating the Empirical Fisher Information Matrix [Morwani et al., 2024]. To address this initialization sensitivity, we implemented a preconditioner warmup phase specifically for Shampoo. Following the methodology in Ren and Goldfarb [2021], we dedicated the first epoch exclusively to accumulating statistics for the preconditioner matrices without updating model parameters. This approach yields a more robust estimation of the preconditioner, which significantly improves the stability of Shampoo during subsequent training iterations. Figure 4 depicts the training loss and validation loss for training NanoGPT with Shampoo, DASGO, Muon, ASGO and Adam-W.

Examining both test and training loss curves, ASGO consistently outperforms Shampoo despite requiring only half the memory consumption and computational effort, which highlights ASGO's practical advantages for training language models. Furthermore, ASGO and Muon achieve the lowest final training and test losses, outperforming all other methods. This consistent performance between ASGO and Muon aligns well with our discussions in Section 6. Moreover, The lightweight DASGO optimizer achieves competitive results against AdamW in both training and test loss metrics. DASGO demonstrates particularly strong performance during the initial training phase (first 200 training steps, which is about the first two epochs), but eventually exhibits a performance gap with ASGO. We discuss this performance gap in Section 7.1.

**Hyperparameter Tuning**    For consistency across all optimization methods, we employ the OneCycleLR learning rate schedule, which has been shown to provide stable convergence properties in deep learning tasks. To ensure optimal performance for each optimizer, we conducted a comprehensive HP search using random search [Bergstra and Bengio, 2012, Choi et al., 2019] within the following predefined ranges: initial lr: $[10^{-5}, 10^{-1}]$; $\beta_1$: $[0.7, 0.99]$; $\beta_2$: $[0.7, 0.99]$; and selected the warmup factor from the set $[0.1, 0.15, 0.2, 0.25, 0.3]$. For the optimizers Shampoo and ASGO, we selected

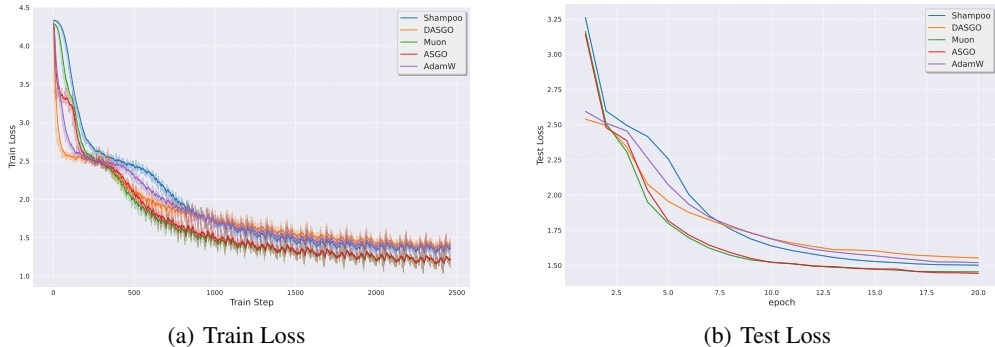

(a) Train Loss

(b) Test Loss

Figure 4: Train Loss and Test Loss on the NanoGPT and Shakespeare Character Dataset

the update frequency $\tau$ from the set $[5, 10, 15]$ and for Shampoo, we selected its IO as either $-\frac{1}{2}$ or $-\frac{1}{4}$. HPs were selected based on validation loss after 20 epochs of training. For each optimizer, we performed a random search for 16 hours to find the optimal HP combination. These best HP choices for training NanoGPT are given in Table 5.

Table 5: Hyperparameter selection for NanoGPT experiment

| Optimizer | learning rate | $\beta_1$ | $\beta_2$ | warmup factor | update frequency | inverse order |
|-----------|---------------|-----------|-----------|---------------|------------------|---------------|
| Muon | 0.00349 | 0.9881 | N/A | 0.3 | N/A | N/A |
| AdamW | 0.00450 | 0.9332 | 0.9528 | 0.2 | N/A | N/A |
| DASGO | 0.060 | 0.9584 | 0.9435 | 0.2 | N/A | N/A |
| Shampoo | 0.00593 | 0.9402 | 0.9760 | 0.3 | 15 | $-\frac{1}{4}$ |
| ASGO | 0.01470 | 0.9541 | 0.8487 | 0.3 | 15 | N/A |

**Ablation Study: Impact of Special Query/Key Design**    In Appendix B, we highlighted the computational benefits of specialized processing for Query and Key matrices in Transformer architectures. To validate these theoretical advantages empirically, we conducted an ablation study comparing the performance and training stability of ASGO and Muon on pretraining NanoGPT, both with and without the specialized Query-Key processing. We maintained the optimal HP configuration established in Table 5 for all parameters except the learning rate. For each algorithm variant (with and without specialized processing), we randomly sampled learning rates from a log-uniform distribution ranging from $10^{-5}$ to $10^{-1}$. Each configuration was trained for 5 epochs on the NanoGPT model, after which we recorded the validation loss. This approach allowed us to evaluate both performance and robustness to learning rate selection. Figure 5 presents the distribution of validation losses after 5 epochs for both algorithms. The violin plots[3] reveal several key insights: (1) For ASGO (Figure 5a), the benefits of specialized Query-Key processing are substantial. The specialized implementation demonstrates a narrower, lower distribution of validation losses (centered around 2.5), indicating greater stability across different learning rates. In contrast, the vanilla implementation shows a long-tailed distribution extending beyond loss values of 6.0, with several outliers representing training instability at certain learning rates. (2) For Muon (Figure 5b), both implementations demonstrate nearly identical performance distributions, with their median validation losses and distribution shapes showing minimal differences. However, the specialized implementation achieves this comparable performance while reducing memory requirements and computational complexity. This represents a clear efficiency advantage – the specialized Query-Key processing effectively provides computational savings with no performance penalty.

### C.2   Pretraining GPT2

**Experimental Setup:**    The model configuration consists of 12 Transformer layers, 12 attention heads, and an embedding dimension of 768. The total number of training steps was 2400, with a batch

---

[3]Violin plot outlines depict empirical probability density; i.e., the width of the shaded area represents the proportion of the data located there. Box plots within a violin plot display the median and inter-quartile range.

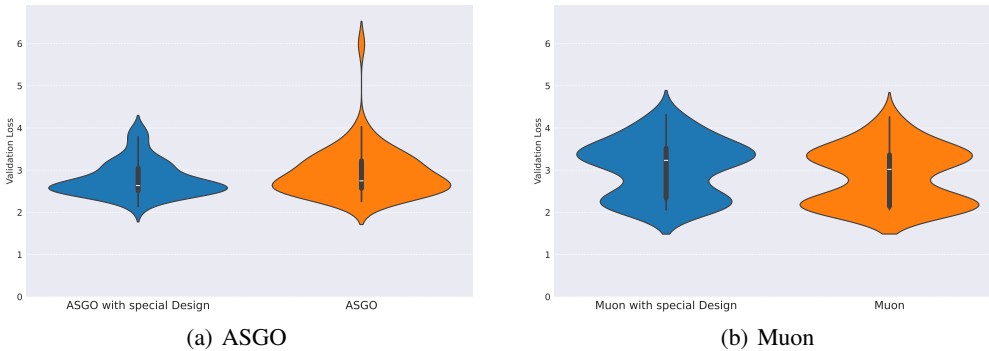

|                    (a) ASGO                    |                    (b) Muon                    |

Figure 5: Distribution of validation losses after 5 epochs with varying learning rates.

size per GPU of 64. The model was trained using Distributed Data Parallel (DDP) on 4 NVIDIA GH200 GPUs, employing gradient accumulation over 8 steps. To ensure a fair comparison across all optimization algorithms, a consistent learning rate schedule was utilized: a linear warmup for the first 200 steps, followed by a cosine annealing decay to a final learning rate of $1 \times 10^{-5}$ for the remainder of the training process. Training was conducted on the OpenWebText dataset with a sequence length of 512 tokens. For Shampoo optimizer, we used the Adam update grafting mentioned in [Shi et al., 2023, Eschenhagen et al., 2025]. For Muon, we used the Adam update RMS norm alignment in [Liu et al., 2025a]. We ran both the Newton-Schultz and the PolarExpress algorithms for 10 steps to compute the inverse square root of $V_t$ in the ASGO optimizer using the following parameters:

- Newton-Schultz: $(a, b, c) = (2, -1.5, 0.5)$

- PolarExpress:
  coeffs_list = [
  (8.28721201814563, -23.595886519098837, 17.300387312530933),
  (4.107059111542203, -2.9478499167379106, 0.5448431082926601),
  (3.9486908534822946 , -2.908902115962949, 0.5518191394370137),
  (3.3184196573706015 , -2.488488024314874, 0.51004894012372),
  (2.300652019954817, -1.6689039845747493, 0.4188073119525673),
  (1.891301407787398, -1.2679958271945868, 0.37680408948524835),
  (1.8750014808534479 , -1.2500016453999487, 0.3750001645474248),
  (1.875, -1.25, 0.375),
  (1.875, -1.25, 0.375),
  (1.875, -1.25, 0.375)]

**Hyperparameter Tuning:** To ensure optimal performance for each optimizer, we conducted a hyperparameter search using grid search. The search space for learning rate (lr) and 2nd momentum $\beta_2$ (where applicable for the optimizer) included the following values:

- Learning rate (lr): $[1 \times 10^{-5}, 1 \times 10^{-4}, 5 \times 10^{-4}, 1 \times 10^{-3}, 5 \times 10^{-3}, 1 \times 10^{-2}, 5 \times 10^{-2}, 0.1]$

- $\beta_2$: [0.7,0.8,0.9,0.95,0.99]

Optimal hyperparameters were selected based on the validation loss achieved after 1000 training steps. The best hyperparameter choices for pretraining GPT-2 are presented in Table 6.

Table 6: Optimal hyperparameter selection for pretraining GPT-2.

| Optimizer | Learning Rate | $\beta_1$ | $\beta_2$ | Weight Decay | Damping $\epsilon$ | Update Freq. ($\tau$) |
|---|---|---|---|---|---|---|
| Muon | 0.01 | 0.9 | N/A | 0.1 | N/A | N/A |
| AdamW | 0.005 | 0.9 | 0.99 | 0.1 | $1 \times 10^{-8}$ | N/A |
| Shampoo | 0.005 | 0.9 | 0.99 | 0.1 | $1 \times 10^{-10}$ | 1 |
| DASGO | 0.01 | 0.9 | 0.9 | 0.1 | $1 \times 10^{-8}$ | N/A |
| ASGO | 0.01 | 0.9 | 0.8 | 0.1 | $1 \times 10^{-10}$ | 1 |

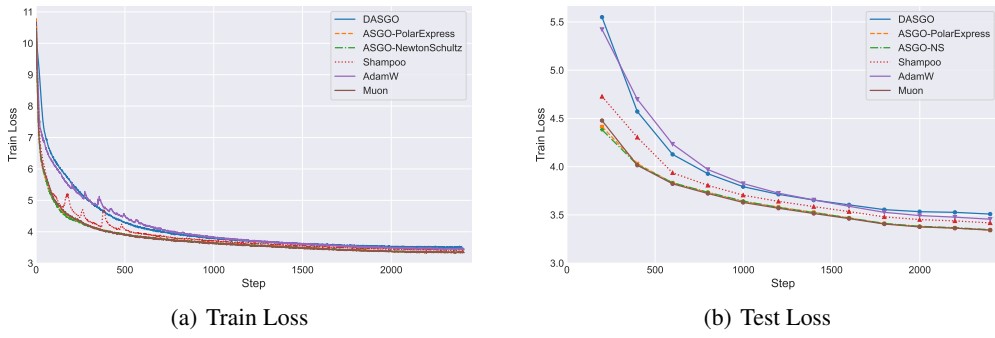

(a) Train Loss           (b) Test Loss

Figure 6: Pretraining Performance on GPT2

## C.3 Finetune GPT2-Large

**Experimental Setup:** We fine-tuned the GPT-2 Large model on the WikiText-2 dataset. We utilized the GPT-2 Large model, which comprises approximately 774 million parameters. This model is a transformer-based language model pretrained on English text using a Causal Language Modeling (CLM) objective, initially developed by OpenAI and was accessed via the Hugging Face Hub. The fine-tuning was performed on the WikiText-2 dataset, a standard benchmark for evaluating language model performance. The dataset was loaded directly using standard library functions. We investigated two distinct fine-tuning objectives:

- Causal Language Modeling (CLM): The conventional autoregressive language modeling task.

- Fill-in-the-Middle (FIM): Adopting the methodology from Bavarian et al. (2022). This objective trains the model to infill masked text spans within a document.

All models were fine-tuned for a total of 2 epochs. For the learning rate, we employed a cosine annealing decay schedule over the course of these 2 epochs, with the learning rate decaying to 0 by the end of training. No warmup phase was used for the fine-tuning learning rate schedule. The fine-tuning process used a batch size of 16 and a sequence length of 128 tokens. For each optimizer evaluated (AdamW, Muon, ASGO, and DASGO), all hyperparameters, with the exception of the learning rate, were maintained at the same values used in our GPT-2 pretraining experiments (as detailed in Table 6).

**Hyperparameter Tuning:** For the fine-tuning experiments on WikiText-2, our hyperparameter tuning focused exclusively on identifying the optimal learning rate for each optimizer under both the CLM and FIM objectives. We performed a grid search over a predefined set of learning rate candidates: $[1 \times 10^{-6}, 5 \times 10^{-5}, 1 \times 10^{-4}, 5 \times 10^{-4}, 1 \times 10^{-3}, 5 \times 10^{-3}]$. The optimal learning rate for each optimizer and fine-tuning objective combination was determined by selecting the learning rate that yielded the lowest validation perplexity on the WikiText-2 validation set after the full 2 epochs of fine-tuning. Table 7 shows the optimal learning rate.

Table 7: Optimal Learning rate from Finetuning GPT2-Large

|  | AdamW | Muon | ASGO | DASGO |
|---|---|---|---|---|
| CLM | $5 \times 10^{-5}$ | $5 \times 10^{-4}$ | $1 \times 10^{-3}$ | $1 \times 10^{-3}$ |
| FIM | $1 \times 10^{-4}$ | $5 \times 10^{-4}$ | $1 \times 10^{-3}$ | $1 \times 10^{-3}$ |

Table 8 presents the average perplexity results obtained after fine-tuning the GPT2-Large (774M) model for 2 epochs on the WikiText-2 dataset. The $\pm$ indicates the 95% confidence intervals of the mean perplexity value over these 5 runs, starting from different random seeds.

**Damping parameter $\epsilon$ sensitivity study:** In this section, we discuss the practical differences between ASGO and Muon. Theoretically, as discussed in Section 6, ASGO can be viewed as an adaptive extension of Muon. Specifically, ASGO's update rule degenerates to that of Muon in the absence of momentum and adaptive scaling (i.e., when $\beta_1 = 0$ and $\beta_2 = 0$).

Table 8: Finetuning GPT2-Large Perplexity

|     | AdamW | Muon | ASGO | DASGO |
|-----|-------|------|------|-------|
| CLM | $14.010 \pm 0.016$ | $13.912 \pm 0.003$ | $13.879 \pm 0.010$ | $13.841 \pm 0.019$ |
| FIM | $17.458 \pm 2.254$ | $15.933 \pm 0.102$ | $15.661 \pm 0.097$ | $15.451 \pm 0.072$ |

Moreover, if we modify the formulas for $V_t$ and $W_{t+1}$ in ASGO (Algorithm 2) to:

$$
\begin{aligned}
&\text{if } m < n \text{ then } V_t = \beta_2 V_{t-1} + (1 - \beta_2) M_t M_t^\top \\
&\text{else } V_t = \beta_2 V_{t-1} + (1 - \beta_2) M_t^\top M_t \\
&\text{if } m < n \text{ then } W_{t+1} = W_t - \eta_t (V_t + \epsilon I)^{-\frac{1}{2}} M_t \\
&\text{else } W_{t+1} = W_t - \eta_t M_t (V_t + \epsilon I)^{-\frac{1}{2}}
\end{aligned}
\tag{4}
$$

and set $\beta_2 = 0$ ASGO becomes identical to Muon, even with momentum. However, in practical training implementations, a subtle yet crucial difference exists between ASGO and Muon. This distinction arises because ASGO applies an additional preconditioner with $\epsilon$ damping to the momentum term, a step not present in the standard Muon update.

To precisely study these differences, we set up an experiment using the Muon optimizer as the background optimizer to train a GPT2-model on the NanoGPT dataset for 1000 steps. At each step, we computed and compared the Muon and ASGO update directions $\Delta W$ in the embedding, query, key, value, and MLP layers. Specifically, we computed:

- **Standard Muon Update Direction (SVD-based)**: $\Delta W_{\text{SVD}}^{\text{Muon}} = U_M V_M^T$, where $U_M$ and $V_M$ are the left and right singular vectors obtained from the first-order momentum $M$.

- **Modified ASGO Update Direction (SVD-based, $\epsilon = 0$)**: $\Delta W_{\text{SVD}, \epsilon=0}^{\text{ASGO}} = V_t^{-\frac{1}{2}} M_t$, where $V_t^{-\frac{1}{2}}$ is computed via standard Singular Value Decomposition (SVD).

- **Modified ASGO Update Direction (SVD-based, $\epsilon = 10^{-8}$)**: $\Delta W_{\text{SVD}, \epsilon=10^{-8}}^{\text{ASGO}} = (V_t + 10^{-8} I)^{-\frac{1}{2}} M_t$, where $(V_t + 10^{-8} I)^{-\frac{1}{2}}$ is computed via standard SVD.

- **Modified ASGO Update Direction (Coupled Newton, $\epsilon = 10^{-8}$)**: $\Delta W_{\text{CN}, \epsilon=10^{-8}}^{\text{ASGO}} = (V_t + 10^{-8} I)^{-\frac{1}{2}} M_t$, where $(V_t + 10^{-8} I)^{-\frac{1}{2}}$ is computed using the Coupled Newton algorithm for 50 iteration steps. Higham [2008a], Shi et al. [2023]

- **Modified ASGO Update Direction (Newton-Schulz, $\epsilon = 10^{-8}$)**: $\Delta W_{\text{NS}, \epsilon=10^{-8}}^{\text{ASGO}} = (V_t + 10^{-8} I)^{-\frac{1}{2}} M_t$, where $(V_t + 10^{-8} I)^{-\frac{1}{2}}$ is computed using the Newton-Schulz algorithm for 50 iteration steps.[4] Higham [2008a], Bernstein and Newhouse [2024b], Jordan et al. [2024]

We then computed the cosine similarity[5] between each of the four modified ASGO update directions and the standard Muon update direction $\Delta W_{\text{SVD}}^{\text{Muon}}$, and plotted these similarities in Figure 7.

Unlike AdamW, ASGO exhibits sensitivity to the choice of the $\epsilon$ damping parameter. As depicted in Figure 7(a), when no damping term is used ($\epsilon = 0$), ASGO's update direction, though exhibiting some instability, largely reconstructs Muon's update direction, with cosine similarities generally exceeding 0.8, particularly for the MLP and Embedding layers, as training progresses. However, the introduction of a small damping hyperparameter ($\epsilon = 10^{-8}$) significantly alters these similarities. Figure 7(b) shows that even with precise SVD computation, the similarities for layers like Value and Key only reach approximately 0.7 gradually. This suggests that the small $\epsilon$ value can introduce and accumulate errors during training, potentially leading to a degradation in optimizer performance, a behavior not typically observed in other common adaptive methods like AdamW. This phenomenon highlights the critical importance of small singular values in the preconditioning process, as applying a damping term can disproportionately affect them, thereby altering the matrix multiplication and

---

[4]The Newton-Schulz algorithm is derived from Algorithm 6.35 on Page 153 in Higham [2008a]. We utilized the quintic version with parameters $(a, b, c) = (2, -1.5, 0.5)$ as described in Jordan et al. [2024].

[5]$\text{CosineSimilarity}(A, B) = \frac{\text{Tr}(A^T B)}{\|A\|_F \|B\|_F}$

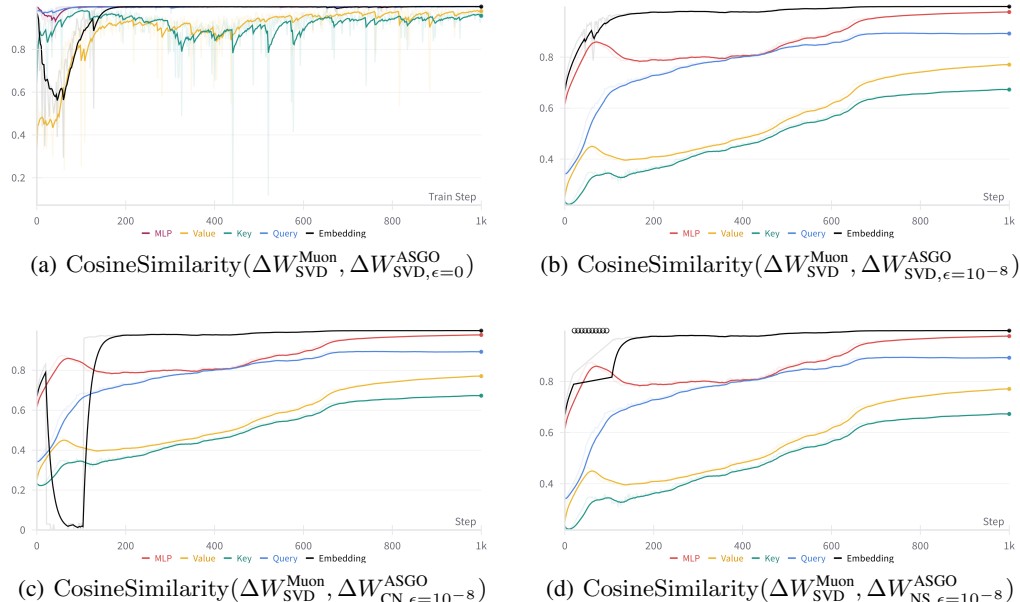

Figure 7: Comparison of Cosine Similarities (CSs) between the Muon update direction ($\Delta W_{\text{SVD}}^{\text{Muon}}$) and various modified ASGO update directions in different layers (MLP, Value, Key, Query, Embedding) during 1000 steps of GPT2 pretraining. Subfigures (a) and (b) plot CSs for SVD-based ASGO update directions with $\epsilon = 0$ and $\epsilon = 10^{-8}$, respectively. Subfigure (c) and (d) plot CSs for Coupled Newton-based and Newton-Schulz-based, respectively, ASGO update directions with $\epsilon = 10^{-8}$.

amplifying errors. To circumvent this sensitivity, we opted to use an $\epsilon$ value of 0 in our large-scale GPT2 pretraining experiments described in Section 7.1. In this setting, we directly compute the SVD of the preconditioning matrix and its inverse square root. The results presented in Figure 6 indeed demonstrate that ASGO achieves performance highly similar to Muon.

In Figure 7(c) and (d), we explore the use of more computationally efficient methods, Coupled Newton (CN) and Newton-Schulz (NS), respectively, to approximately compute $V_t^{-\frac{1}{2}}$ as alternatives to SVD. The results show that, aside from some numerical stability issues observed in the Embedding layer for both CN and NS at certain steps—which is reasonable given that the sparse nature of embedding layers often requiring larger damping parameters—the cosine similarities for the remaining parameters are highly comparable to those obtained with SVD. This demonstrates the feasibility of employing efficient matrix algorithms to replace SVD, thereby enhancing ASGO's computational efficiency. This direction represents one of our promising avenues for future research.

# D  Auxiliary Lemmas for the Proof

**Lemma 1** (Trace properties). *For arbitrary matrices $A \in \mathbb{R}^{m \times n}$, $B \in \mathbb{R}^{n \times m}$, $X, Y \in \mathbb{R}^{m \times m}$, we have the following basic properties for the trace:*

*1. $\operatorname{tr}(X) = \operatorname{tr}(X^\top)$;*

*2. $\operatorname{tr}(AB) = \operatorname{tr}(BA)$;*

*3. if $X$ is symmetric positive semidefinite, $\operatorname{tr}(X) = \|X\|_* \geq 0$;*

*4. if $X, Y \succeq 0$ and $X$ is symmetric, $\operatorname{tr}(XY) \geq 0$.*

The following lemma notes the operator monotonicity of the power functions, which is a classic result [Löwner, 1934, Ando et al., 2004, Gupta et al., 2018].

**Lemma 2.** *The function $f : x \to x^\alpha$ with $\alpha \in [0, 1]$ is operator-monotone, i.e. if $0 \preceq A \preceq B$, it holds that $A^\alpha \preceq B^\alpha$.*

**Lemma 3.** *For symmetric positive definite matrices $X, Y \in \mathbb{R}^{m \times m}$, it holds that*

$$\mathrm{tr}\left((X+Y)^{\frac{1}{2}}\right) \leq \mathrm{tr}\left(X^{\frac{1}{2}} + Y^{\frac{1}{2}}\right).$$

*Proof.* It holds that

$$
\begin{aligned}
\mathrm{tr}\left((X+Y)^{\frac{1}{2}}\right) &= \mathrm{tr}\left((X+Y)^{-\frac{1}{2}}(X+Y)\right) \\
&= \mathrm{tr}\left((X+Y)^{-\frac{1}{2}}X\right) + \mathrm{tr}\left((X+Y)^{-\frac{1}{2}}Y\right) \\
&\geq \mathrm{tr}\left(X^{\frac{1}{2}}\right) + \mathrm{tr}\left(Y^{\frac{1}{2}}\right),
\end{aligned}
$$

where the last inequality is based on Lemma 2 such that $(X+Y)^{\frac{1}{2}} \succeq X^{\frac{1}{2}}$ and the fact that $A^{-1} \preceq B^{-1}$ if $A \succeq B$ for symmetric positive definite matrices $A, B$. Further based on Lemma 1, we can finish the proof. $\square$

**Lemma 4.** *For a symmetric positive semidefinite matrix $X \in \mathbb{R}^{m \times m}$, it holds that*

$$\mathrm{tr}\left(X^{\frac{1}{2}}\right) \leq \sum_{j=1}^{m} \sqrt{[X]_{j,j}}.$$

*Proof.* The inequality is equivalent to that

$$\sum_{i=1}^{m} \sqrt{\lambda_i(X)} \leq \sum_{i=1}^{m} \sqrt{[X]_{i,i}},$$

where $\lambda_i(X)$ denotes the $i$-th largest eigenvalue of $X$ (the same as singular values for real symmetric positive semidefinite matrices). Firstly, we have the fact that the eigenvalues $\{\lambda_i(X)\}_{i=1}^{m}$ majorize the diagonal entries $\{[X]_{i,i}\}_{i=1}^{m}$, i.e. for all $1 \leq l < m$,

$$\sum_{i=1}^{l} \lambda_i(X) \geq \sum_{i=1}^{l} [X]_{i,i}, \quad \text{and} \quad \sum_{i=1}^{m} \lambda_i(X) = \sum_{i=1}^{m} [X]_{i,i}.$$

Also, we know $g : \{x_i\}_{i=1}^{m} \to \sum_{i=1}^{m} x_i^{\frac{1}{2}}$ is a Schur-concave operator. Then we obtain the equality based on the Schur-Horn theorem, which implies that for a Schur-concave operator $g$, if sequence $\{x_i\}_{i=1}^{m}$ majorizes $\{y_i\}_{i=1}^{m}$, then $g(\{x_i\}_{i=1}^{m}) \leq g(\{y_i\}_{i=1}^{m})$. $\square$

**Lemma 5.** *$\|\cdot\|_L$ is a norm, i.e., it satisfies the basic properties of norms. Its dual norm is $\|\cdot\|_{L^{-1}}$.*

*Proof.* Denote $X = [x_1, \ldots, x_n] \in \mathbb{R}^{m \times n}$, where each $x_i \in \mathbb{R}^{1 \times m}$. Then we have

$$\mathrm{tr}\left(X^\top L X\right) = \sum_{i=1}^{n} x_i^\top L x_i = \begin{bmatrix} x_1 \\ \cdots \\ x_n \end{bmatrix}^\top \begin{bmatrix} L & & \\ & L & \\ & & \cdots & \\ & & & L \end{bmatrix} \begin{bmatrix} x_1 \\ \cdots \\ x_{n,} \end{bmatrix},$$

which is a norm for the space $\mathbb{R}^{m \times n}$ whose dual norm is $\|\cdot\|_{L^{-1}}$. This concludes the proof. $\square$

**Lemma 6.** *Assume a non-negative sequence $\{x_j\}_{j=1}^{n}$ and a positive sequence $\{s_j\}_{j=1}^{n}$ with $S = \sum_{i=1}^{n} s_j$, it holds that*

$$\frac{1}{S} \sum_{j=1}^{n} x_j \leq \sqrt{\frac{1}{S} \sum_{j=1}^{n} \frac{x_j^2}{s_j}}. \tag{5}$$

*The inequality holds as an equality if and only if for all $i = 1, \cdots, n$ and $j = 1, \cdots, n$,*

$$\frac{x_i}{s_i} = \frac{x_j}{s_j}.$$

*Proof.* A proof can be found in Lemma G.5 of the ICLR version of Liu et al. [2024]. $\square$

# E   Proof of Nonsmooth Convergence of ASGO

*Proof of Theorem 1.* Denote $\Delta W_t \triangleq W_t - W_*$ where $W_*$ is an optimum of Problem (1). From the algorithm update, we have

$$\Delta W_{t+1}^\top \Lambda_t \Delta W_{t+1} = \Delta W_t^\top \Lambda_t \Delta W_t - \eta_t \left( G_t^\top \Delta W_t + \Delta W_t^\top G_t \right) + \eta_t^2 G_t^\top \Lambda_t^{-1} G_t.$$

Then, by taking trace of the above equality and rearranging, we can obtain that

$$2\eta_t \mathrm{tr} \left( G_t^\top \Delta W_t \right) = \mathrm{tr} \left( \Delta W_t^\top \Lambda_t \Delta W_t - \Delta W_{t+1}^\top \Lambda_t \Delta W_{t+1} \right) + \eta_t^2 \mathrm{tr} \left( G_t^\top \Lambda_t^{-1} G_t \right).$$

By convexity, we know

$$\mathbb{E} \left[ \mathrm{tr} \left( G_t^\top \Delta W_t \right) \right] = \mathbb{E} \left[ \langle \nabla f(W_t), \Delta W_t \rangle \right] \geq \mathbb{E}[f(W_t)] - f(W_*).$$

Then combining these and taking summation over $t$ and taking $\eta_t \equiv \eta$, we have

$$2 \sum_{t=0}^{T-1} \mathbb{E}[f(W_t)] - f(W_*)$$

$$\leq \frac{1}{\eta} \mathbb{E} \left[ \sum_{t=0}^{T-1} \mathrm{tr} \left( \Delta W_t^\top \Lambda_t \Delta W_t - \Delta W_{t+1}^\top \Lambda_t \Delta W_{t+1} \right) \right] + \eta \mathbb{E} \left[ \sum_{t=0}^{T-1} \mathrm{tr} \left( G_t^\top \Lambda_t^{-1} G_t \right) \right]$$

$$= \frac{1}{\eta} \mathbb{E} \left[ \sum_{t=0}^{T-1} \mathrm{tr} \left( \Delta W_t^\top \Lambda_t \Delta W_t - \Delta W_t^\top \Lambda_{t-1} \Delta W_t \right) \right] + \frac{1}{\eta} \mathrm{tr} \left( \Delta W_0^\top \Lambda_{-1} \Delta W_0 \right) - \frac{1}{\eta} \mathbb{E} \left[ \mathrm{tr} \left( \Delta W_T^\top \Lambda_{T-1} \Delta W_T \right) \right]$$

$$+ \eta \mathbb{E} \left[ \sum_{t=0}^{T-1} \mathrm{tr} \left( G_t^\top \Lambda_t^{-1} G_t \right) \right]$$

$$\leq \frac{1}{\eta} \mathrm{tr} \left( \Delta W_0^\top \Lambda_{-1} \Delta W_0 \right) + \frac{1}{\eta} \mathbb{E} \left[ \sum_{t=0}^{T-1} \mathrm{tr} \left( \Delta W_t^\top (\Lambda_t - \Lambda_{t-1}) \Delta W_t \right) \right] + \eta \mathbb{E} \left[ \sum_{t=0}^{T-1} \mathrm{tr} \left( G_t^\top \Lambda_t^{-1} G_t \right) \right], \tag{6}$$

where we note $\Lambda_{-1} = \epsilon I_m$. Thus the first term on the RHS of (6) can be bounded by $\epsilon$ as

$$\frac{1}{\eta} \mathrm{tr} \left( \Delta W_0^\top \Lambda_{-1} \Delta W_0 \right) = \frac{\epsilon}{\eta} \left\| \Delta W_0 \right\|_F^2 \leq \frac{\epsilon D_F^2}{\eta}. \tag{7}$$

We then deal with the second and third terms separately. For the second term, we have

$$\sum_{t=0}^{T-1} \mathrm{tr} \left( \Delta W_t^\top (\Lambda_t - \Lambda_{t-1}) \Delta W_t \right) = \sum_{t=0}^{T-1} \mathrm{tr} \left( \Delta W_t \Delta W_t^\top (\Lambda_t - \Lambda_{t-1}) \right)$$

$$= \sum_{t=0}^{T-1} \left\langle \Delta W_t \Delta W_t^\top, \Lambda_t - \Lambda_{t-1} \right\rangle$$

$$\leq \sum_{t=0}^{T-1} \left\| \Delta W_t \Delta W_t^\top \right\|_{\mathrm{op}} \left\| \Lambda_t - \Lambda_{t-1} \right\|_*$$

$$= \sum_{t=0}^{T-1} \left\| \Delta W_t \right\|_{\mathrm{op}}^2 \mathrm{tr} \left( \Lambda_t - \Lambda_{t-1} \right) \leq D_{\mathrm{op}}^2 \mathrm{tr} \left( \Lambda_{T-1} - \mathrm{tr} \left( \Lambda_{-1} \right) \right). \tag{8}$$

Note that the first and last equalities are based on the properties of the trace, (see Lemma 1). The first inequality is based on the duality of the $\|\cdot\|_{\mathrm{op}}$ and $\|\cdot\|_*$ norms. The third equality relies on the positive semidefiniteness of $\Lambda_t - \Lambda_{t-1}$ for all $t$ based on Lemma 2, since $\Lambda_t^2 - \Lambda_{t-1}^2 = G_t G_t^\top \succeq 0$.

For the third term of (6), we have

$$\mathrm{tr} \left( G_t^\top \Lambda_t^{-1} G_t \right) = \mathrm{tr} \left( G_t G_t^\top \Lambda_t^{-1} \right) = \mathrm{tr} \left( (\Lambda_t^2 - \Lambda_{t-1}^2) \Lambda_t^{-1} \right)$$

$$= \mathrm{tr} \left( \left( (\Lambda_t - \Lambda_{t-1}) \cdot 2\Lambda_t - (\Lambda_t - \Lambda_{t-1})^2 + \Lambda_{t-1}\Lambda_t - \Lambda_t\Lambda_{t-1} \right) \Lambda_t^{-1} \right)$$

$$= 2\mathrm{tr} \left( \Lambda_t - \Lambda_{t-1} \right) - \mathrm{tr} \left( (\Lambda_t - \Lambda_{t-1})^2 \Lambda_t^{-1} \right) + \mathrm{tr} \left( (\Lambda_{t-1}\Lambda_t - \Lambda_t\Lambda_{t-1}) \Lambda_t^{-1} \right) \tag{9}$$

$$\leq 2\mathrm{tr} \left( \Lambda_t - \Lambda_{t-1} \right),$$

where the last inequality follows from the fact that

$$\text{tr}\left((\Lambda_t - \Lambda_{t-1})^2 \Lambda_t^{-1}\right) = \text{tr}\left(\left[(\Lambda_t - \Lambda_{t-1})\Lambda_t^{-\frac{1}{2}}\right]\left[(\Lambda_t - \Lambda_{t-1})\Lambda_t^{-\frac{1}{2}}\right]^\top\right) \geq 0$$

and

$$\text{tr}\left((\Lambda_{t-1}\Lambda_t - \Lambda_t \Lambda_{t-1})\Lambda_t^{-1}\right) = \text{tr}\left(\Lambda_{t-1}\right) - \text{tr}\left(\Lambda_t \Lambda_{t-1}\Lambda_t^{-1}\right) = 0,$$

which are based on the positive definiteness of $\Lambda_t$ and the properties of the trace (Lemma 1). Therefore, by substituting (7), (8), and (9) into (6), we have

$$2 \sum_{t=0}^{T-1} \mathbb{E}[f(W_t)] - f(W_*)$$

$$\leq \frac{1}{\eta}\text{tr}\left(\Delta W_0^\top \Lambda_{-1}\Delta W_0\right) + \frac{1}{\eta}\mathbb{E}\left[\sum_{t=0}^{T-1}\text{tr}\left(\Delta W_t^\top (\Lambda_t - \Lambda_{t-1})\Delta W_t\right)\right] + \eta\mathbb{E}\left[\sum_{t=0}^{T-1}\text{tr}\left(G_t^\top \Lambda_t^{-1}G_t\right)\right]$$

$$\leq \frac{\epsilon D_\text{F}^2}{\eta} + \left(\frac{D_\text{op}^2}{\eta} + 2\eta\right)\mathbb{E}\left[\text{tr}\left(\Lambda_{T-1} - \Lambda_{-1}\right)\right]$$

$$= \left(\frac{D_\text{op}^2}{\eta} + 2\eta\right)\mathbb{E}\left[\text{tr}\left(\left(\sum_{t=0}^{T-1} G_t G_t^\top\right)^{\frac{1}{2}}\right)\right] + \frac{\epsilon D_\text{F}^2}{\eta}.$$

Taking $\eta = D_\text{op}$ completes the proof. $\qquad\square$

Based on Theorem 1 and Lemma 8, we can also prove Corollary 2.

*Proof of Corollary 2.* Based on the results in Theorem 1, if we additionally have

$$\mathbb{E}\left[G_t G_t^\top\right] \preceq Q^2,$$

then using Lemma 8, we can obtain that

$$\mathbb{E}\left[\left\|\left(\sum_{t=0}^{T-1} G_t G_t^\top\right)^{\frac{1}{2}}\right\|_*\right] \overset{(10)}{\leq} \mathbb{E}\left[\sqrt{\|Q\|_* \text{tr}\left(\left(\sum_{t=0}^{T-1} G_t G_t^\top\right)^{\frac{1}{2}} Q^{-1}\left(\sum_{t=0}^{T-1} G_t G_t^\top\right)^{\frac{1}{2}}\right)}\right]$$

$$= \mathbb{E}\left[\sqrt{\|Q\|_* \text{tr}\left(\sum_{t=0}^{T-1} G_t G_t^\top Q^{-1}\right)}\right]$$

$$\leq \sqrt{\|Q\|_* \sum_{t=0}^{T-1}\text{tr}\left(\mathbb{E}\left[G_t G_t^\top\right] Q^{-1}\right)}$$

$$\leq \sqrt{\|Q\|_* \sum_{t=0}^{T-1}\|Q\|_*} = \sqrt{T}\|Q\|_*,$$

where the first equality is based on Lemma 1 and the second inequality is based on the fact that $g(x) = \sqrt{x}$ is concave for $x \geq 0$. This concludes the proof. $\qquad\square$

We also provide the following proof for the comparison between Theorem 1 and the convergence rates of Shampoo and full-matrix AdaGrad.

*Proof of Comparison with Shampoo and full-matrix AdaGrad.* We list the convergence rates here.

$$\text{Full-Matrix AdaGrad:}\quad \mathcal{O}\left(D_\text{F}\sum_{j=1}^m \sum_{i=1}^n \sqrt{\sum_{t=0}^{T-1}[G_t]_{i,j}^2}\right)$$

$$\text{Shampoo:} \quad \mathcal{O}\left(\sqrt{r_G}D_{\mathrm{F}} \cdot \mathrm{tr}\left(\left(\sum_{t=0}^{T-1} G_t G_t^\top\right)^{\frac{1}{4}}\right) \cdot \mathrm{tr}\left(\left(\sum_{t=0}^{T-1} G_t^\top G_t\right)^{\frac{1}{4}}\right)\right)$$

$$\text{ASGO:} \quad \mathcal{O}\left(D_{\mathrm{op}} \cdot \mathrm{tr}\left(\left(\sum_{t=0}^{T-1} G_t G_t^\top\right)^{\frac{1}{2}}\right)\right) \leq \mathcal{O}\left(D_{\mathrm{op}} \cdot \sum_{j=1}^{m}\sqrt{\sum_{i=1}^{n}\sum_{t=0}^{T-1}[G_t]_{i,j}^2}\right).$$

The last inequality for ASGO is based on Lemma 4. We first compare ASGO and full-matrix AdaGrad:

$$\mathrm{tr}\left(\left(\sum_{t=0}^{T-1} G_t G_t^\top\right)^{\frac{1}{2}}\right) \leq \sum_{j=1}^{m}\sqrt{\left[\sum_{t=0}^{T-1} G_t G_t^\top\right]_{j,j}} = \sum_{j=1}^{m}\sqrt{\sum_{i=1}^{n}\sum_{t=0}^{T-1}[G_t]_{i,j}^2} \leq \sum_{j=1}^{m}\sum_{i=1}^{n}\sqrt{\sum_{t=0}^{T-1}[G_t]_{i,j}^2},$$

where the first inequality is based on Lemma 4 and the second inequality is simply a fact that for any vector $x$, we have $\|x\|_2 \geq \|x\|_1$. Thus we prove that the proven convergence rate of ASGO is at least $D_{\mathrm{F}}/D_{\mathrm{op}}$ times faster than full-matrix AdaGrad.

Then we compare ASGO and Shampoo. We have

$$\mathrm{tr}\left(\left(\sum_{t=0}^{T-1} G_t G_t^\top\right)^{\frac{1}{2}}\right) = \left\langle\left(\sum_{t=0}^{T-1} G_t G_t^\top\right)^{\frac{1}{4}}, \left(\sum_{t=0}^{T-1} G_t G_t^\top\right)^{\frac{1}{4}}\right\rangle$$

$$\leq \mathrm{tr}\left(\left(\sum_{t=0}^{T-1} G_t G_t^\top\right)^{\frac{1}{4}}\right) \cdot \left\|\left(\sum_{t=0}^{T-1} G_t G_t^\top\right)^{\frac{1}{4}}\right\|_{\mathrm{op}}$$

$$= \mathrm{tr}\left(\left(\sum_{t=0}^{T-1} G_t G_t^\top\right)^{\frac{1}{4}}\right) \cdot \left\|\sum_{t=0}^{T-1} G_t G_t^\top\right\|_{\mathrm{op}}^{\frac{1}{4}},$$

where the inequality is based on the fact that $\|\cdot\|_{\mathrm{op}}$ and $\|\cdot\|_*$ are dual norms. Then we have

$$\left\|\sum_{t=0}^{T-1} G_t G_t^\top\right\|_{\mathrm{op}}^{\frac{1}{4}} \leq \left(\sum_{t=0}^{T-1}\|G_t G_t^\top\|_{\mathrm{op}}\right)^{\frac{1}{4}} = \left(\sum_{t=0}^{T-1}\|G_t^\top G_t\|_{\mathrm{op}}\right)^{\frac{1}{4}}$$

$$\leq \left(\mathrm{tr}\left(\sum_{t=0}^{T-1} G_t^\top G_t\right)\right)^{\frac{1}{4}} \leq \mathrm{tr}\left(\left(\sum_{t=0}^{T-1} G_t^\top G_t\right)^{\frac{1}{4}}\right),$$

where the first inequality is based on that $\|\cdot\|_{\mathrm{op}}$ is a norm and the second inequality is based on the fact that $\mathrm{tr}(X) \geq \|X\|_{\mathrm{op}}$ for symmetric positive semidefinite $X$, and the third inequality is based on the fact that $(\mathrm{tr}(X))^{\frac{1}{4}} \leq \mathrm{tr}\left(X^{\frac{1}{4}}\right)$ because if we denote $\sigma_j$ as the $j$-th singular value of $X$, we have

$$(\mathrm{tr}(X))^{\frac{1}{4}} = \left(\sum_{j=1}^{r}\sigma_j\right)^{\frac{1}{4}} \leq \sum_{j=1}^{r}\sigma_j^{\frac{1}{4}}.$$

Thus we can conclude that

$$\mathrm{tr}\left(\left(\sum_{t=0}^{T-1} G_t G_t^\top\right)^{\frac{1}{2}}\right) \leq \mathrm{tr}\left(\left(\sum_{t=0}^{T-1} G_t G_t^\top\right)^{\frac{1}{4}}\right) \cdot \mathrm{tr}\left(\left(\sum_{t=0}^{T-1} G_t^\top G_t\right)^{\frac{1}{4}}\right).$$

Therefore, the proven rate of ASGO is at least $\sqrt{r_G}D_{\mathrm{F}}/D_{\mathrm{op}}$ times faster than Shampoo. $\qquad\square$

# F  Proof of Smooth Convergence of ASGO

The starting point of the smooth analysis is Theorem 1. For notation simplicity, we denote $N_t \triangleq G_t - \nabla f(W_t)$ and $\nabla f_t \triangleq \nabla f(W_t)$ in this section.

**Lemma 7** (Separate Gradient and Noise). *Under the same settings as Theorem 1, it holds that*

$$\frac{1}{T}\sum_{t=0}^{T-1}\mathbb{E}[f(W_t)-f(W_*)] \leq \frac{D_{\mathrm{op}}}{T}\mathbb{E}\left[\left\|\left(2\sum_{t=0}^{T-1}\nabla f_t\nabla f_t^\top\right)^{\frac{1}{2}}\right\|_*\right] + \frac{D_{\mathrm{op}}}{T}\mathbb{E}\left[\left\|\left(2\sum_{t=0}^{T-1}N_tN_t^\top\right)^{\frac{1}{2}}\right\|_*\right] + \frac{\epsilon D_{\mathrm{F}}^2}{D_{\mathrm{op}}T}.$$

*Proof.* Based on Theorem 1, we have

$$\frac{1}{T}\sum_{t=0}^{T-1}\mathbb{E}[f(W_t)-f(W_*)] \leq \frac{D_{\mathrm{op}}}{T}\mathbb{E}\left[\left\|\left(\sum_{t=0}^{T-1}G_tG_t^\top\right)^{\frac{1}{2}}\right\|_*\right] + \frac{\epsilon D_{\mathrm{F}}^2}{D_{\mathrm{op}}T}$$

$$\leq \frac{D_{\mathrm{op}}}{T}\mathbb{E}\left[\left\|\left(\sum_{t=0}^{T-1}2\nabla f_t\nabla f_t^\top + 2N_tN_t^\top\right)^{\frac{1}{2}}\right\|_*\right] + \frac{\epsilon D_{\mathrm{F}}^2}{D_{\mathrm{op}}T}$$

$$\leq \frac{D_{\mathrm{op}}}{T}\mathbb{E}\left[\left\|\left(2\sum_{t=0}^{T-1}\nabla f_t\nabla f_t^\top\right)^{\frac{1}{2}}\right\|_*\right] + \frac{D_{\mathrm{op}}}{T}\mathbb{E}\left[\left\|\left(2\sum_{t=0}^{T-1}N_tN_t^\top\right)^{\frac{1}{2}}\right\|_*\right] + \frac{\epsilon D_{\mathrm{F}}^2}{D_{\mathrm{op}}T}.$$

Here the first inequality comes directly from Theorem 1. The second inequality is based on the fact that for all $A, B \in \mathbb{R}^{m \times n}$ we have

$$2AA^\top + 2BB^\top - (A+B)(A+B)^\top = AA^\top + BB^\top - A^\top B - B^\top A$$
$$= (A-B)(A-B)^\top \succeq 0.$$

The last inequality is based on Lemma 3. □

The following lemma is a key technical lemma for proving the smooth convergence results, which is a generalization to the matrix case of Lemma 6.

**Lemma 8** (An upper bound on $\|\cdot\|_*$). *For a symmetric positive definite matrix $\Lambda \in \mathbb{R}^{m \times m}$ and matrix $G \in \mathbb{R}^{m \times n}$, it holds that*

$$\|G\|_* \leq \sqrt{\|\Lambda\|_* \operatorname{tr}\left(G^\top \Lambda^{-1} G\right)}. \tag{10}$$

*Proof.* We first consider the case $\Lambda = \operatorname{diag}[\lambda_1, \ldots, \lambda_m]$ is a diagonal matrix. We have

$$\|G\|_* = \operatorname{tr}\left((GG^\top)^{\frac{1}{2}}\right) \leq \sum_{j=1}^m \sqrt{[GG^\top]_{j,j}}$$

$$\leq \sqrt{\left(\sum_{j=1}^m \lambda_j\right)\left(\sum_{j=1}^m \frac{[GG^\top]_{j,j}}{\lambda_j}\right)}$$

$$= \sqrt{\|\Lambda\|_* \operatorname{tr}\left(GG^\top \Lambda^{-1}\right)} = \sqrt{\|\Lambda\|_* \operatorname{tr}\left(G^\top \Lambda^{-1} G\right)},$$

where the first inequality is based on Lemma 4 and the second inequality is based on Lemma 6.

Then we prove that this holds for general symmetric positive definite matrix $\Lambda$. Let the singular value decomposition be $\Lambda = U\Sigma U^\top$. Denote $\tilde{G} = U^\top G$. Since $\Sigma \in \mathbb{R}^{m \times m}$ is a diagonal matrix, it holds that

$$\|G\|_* = \left\|\tilde{G}\right\|_* \leq \sqrt{\|\Sigma\|_* \operatorname{tr}\left(\tilde{G}^\top \Sigma^{-1} \tilde{G}\right)} = \sqrt{\|\Lambda\|_* \operatorname{tr}\left(G^\top U\Sigma^{-1}U^\top G\right)}$$

$$= \sqrt{\|\Lambda\|_* \operatorname{tr}\left(G^\top \Lambda^{-1} G\right)},$$

which concludes the proof. □

We also use the following lemma to indicate the reduction of variance by batch size $M$.

**Lemma 9** (Variance reduction by batch size). *Under Assumption 3, we have*

$$\mathbb{E}\left[N_t N_t^\top\right] \preceq \frac{1}{M}V^2,$$

*where we denote $N_t \triangleq G_t - \nabla f(W_t)$ for Algorithm 1.*

*Proof.* For notation simplicity, we denote $N(W_t; \xi) \triangleq \nabla f(W_t; \xi) - \nabla f(W_t)$ and thus

$$N_t = \frac{1}{M}\sum_{\xi \in \mathcal{B}} N(W_t; \xi).$$

Then it holds that

$$\mathbb{E}\left[N_t N_t^\top\right] = \frac{1}{M^2}\mathbb{E}\left[\left(\sum_{\xi \in \mathcal{B}} N(W_t; \xi)\right)\left(\sum_{\zeta \in \mathcal{B}} N(W_t; \zeta)\right)^\top\right]$$

$$= \frac{1}{M^2}\sum_{\xi \in \mathcal{B}}\sum_{\zeta \in \mathcal{B}}\mathbb{E}\left[N(W_t; \xi)N(W_t; \zeta)^\top\right]$$

$$= \frac{1}{M^2}\sum_{\xi \in \mathcal{B}}\mathbb{E}\left[N(W_t; \xi)N(W_t; \xi)^\top\right]$$

$$\preceq \frac{1}{M^2}\sum_{\xi \in \mathcal{B}} V^2 = \frac{1}{M}V^2,$$

where the last equality is based on that $\nabla f(W_t; \xi)$ are mutually independent and $\mathbb{E}[N(W_t; \xi)] = 0$ and the last inequality is based on Assumption 3. $\qquad\square$

Then based on these lemmas, we can prove the smooth results.

*Proof of Theorem 3.* We separately deal with the bias and variance terms, which refer to the first two terms on the RHS of Lemma 7. For the bias part, by plugging in the smoothness matrix $L$ into Lemma 8, we can obtain

$$\mathbb{E}\left[\left\|\left(\sum_{t=0}^{T-1}\nabla f_t \nabla f_t^\top\right)^{\frac{1}{2}}\right\|_*\right] \stackrel{(10)}{\leq} \mathbb{E}\left[\sqrt{\|L\|_* \operatorname{tr}\left(\left(\sum_{t=0}^{T-1}\nabla f_t \nabla f_t^\top\right)^{\frac{1}{2}} L^{-1}\left(\sum_{t=0}^{T-1}\nabla f_t \nabla f_t^\top\right)^{\frac{1}{2}}\right)}\right]$$

$$= \mathbb{E}\left[\sqrt{\|L\|_* \operatorname{tr}\left(\sum_{t=0}^{T-1}\nabla f_t \nabla f_t^\top L^{-1}\right)}\right]$$

$$\leq \sqrt{\|L\|_* \sum_{t=0}^{T-1}\mathbb{E}\left[\|\nabla f_t\|_{L^{-1}}^2\right]} \leq \sqrt{\|L\|_* \sum_{t=0}^{T-1} 2(f(W_t) - f(W_*))},$$

where the equalities are based on Lemma 1 and the second inequality is based on the fact that $g(x) = \sqrt{x}$ is concave for $x \geq 0$. The last inequality is based on the properties of smoothness [Nesterov et al., 2018] and the fact that $\|\cdot\|_{L^{-1}}$ is the dual norm of $\|\cdot\|_L$.

Then, for the variance part, let us first assume $M = 1$ for simplicity, then we have

$$\mathbb{E}\left[\left\|\left(\sum_{t=0}^{T-1} N_t N_t^\top\right)^{\frac{1}{2}}\right\|_*\right] \stackrel{(10)}{\leq} \mathbb{E}\left[\sqrt{\|V\|_* \operatorname{tr}\left(\left(\sum_{t=0}^{T-1} N_t N_t^\top\right)^{\frac{1}{2}} V^{-1}\left(\sum_{t=0}^{T-1} N_t N_t^\top\right)^{\frac{1}{2}}\right)}\right]$$

$$= \mathbb{E}\left[\sqrt{\|V\|_* \operatorname{tr}\left(\sum_{t=0}^{T-1} N_t N_t^\top V^{-1}\right)}\right]$$

$$\leq \sqrt{\|V\|_* \sum_{t=0}^{T-1} \mathrm{tr}\left(\mathbb{E}\left[N_t N_t^\top\right] V^{-1}\right)}$$

$$\leq \sqrt{\|V\|_* \sum_{t=0}^{T-1} \|V\|_*} = \sqrt{T}\|V\|_*,$$

where the equality is based on Lemma 1. and the second inequality is based on the fact that $g(x) = \sqrt{x}$ is concave for $x \geq 0$. Then plugging these in Lemma 7, we have

$$\frac{1}{T}\sum_{t=0}^{T-1}\mathbb{E}[f(W_t) - f(W_*)] \leq \frac{D_{\mathrm{op}}}{T}\mathbb{E}\left[\left\|\left(2\sum_{t=0}^{T-1}\nabla f_t \nabla f_t^\top\right)^{\frac{1}{2}}\right\|_*\right] + \frac{D_{\mathrm{op}}}{T}\mathbb{E}\left[\left\|\left(2\sum_{t=0}^{T-1}N_t N_t^\top\right)^{\frac{1}{2}}\right\|_*\right] + \frac{\epsilon D_{\mathrm{F}}^2}{D_{\mathrm{op}}T}$$

$$\leq \frac{2D_{\mathrm{op}}}{T}\sqrt{\|L\|_* \sum_{t=0}^{T-1}\mathbb{E}[f(W_t) - f(W_*)]} + \frac{\sqrt{2}D_{\mathrm{op}}\|V\|_*}{\sqrt{T}} + \frac{2\epsilon D_{\mathrm{F}}^2}{D_{\mathrm{op}}T}.$$

Thus if we denote $x = \sqrt{\frac{1}{T}\sum_{t=0}^{T-1}\mathbb{E}[f(W_t) - f(W_*)]}$, we can write the inequality as

$$x^2 \leq bx + c,$$

where

$$b = \frac{2D_{\mathrm{op}}\sqrt{\|L\|_*}}{\sqrt{T}}, \quad c = \frac{\sqrt{2}D_{\mathrm{op}}\|V\|_*}{\sqrt{T}} + \frac{\epsilon D_{\mathrm{F}}^2}{D_{\mathrm{op}}T}.$$

Then as $x \geq 0$, we can solve this simple quadratic inequality to obtain that

$$x^2 \leq \frac{1}{4}\left(b + \sqrt{b^2 + 4c}\right)^2 \leq 2b^2 + 2c$$

$$\iff \frac{1}{T}\sum_{t=0}^{T-1}\mathbb{E}[f(W_t) - f(W_*)] \leq \frac{4D_{\mathrm{op}}^2\|L\|_*}{T} + \frac{2\sqrt{2}D_{\mathrm{op}}\|V\|_*}{\sqrt{T}} + \frac{2\epsilon D_{\mathrm{F}}^2}{D_{\mathrm{op}}T},$$

which concludes the proof for $M = 1$. Then, based on Lemma 9 to incorporate batch size $M > 1$, we finish the proof. $\square$

## G  Proof of Section 6

Muon [Jordan et al., 2024] is essentially the following algorithm:

$$B_t = \mu B_{t-1} + G_t$$
$$U_t, S_t, V_t = \text{Compact SVD}(B_t) \tag{11}$$
$$W_{t+1} = W_t - \eta_t U_t V_t^\top,$$

where $G_t$ is the gradient obtained at $W_t$ and $U_t \in \mathbb{R}^{m \times r_t}, V_t \in \mathbb{R}^{r_t \times n}$ are the matrices of the $r_t$ left and right singular vectors corresponding to the non-zero singular values of $B_t$ that are the diagonal components of the diagonal matrix $S_t \in \mathbb{R}^{r_t \times r_t}$ with $r_t$ being the rank of $B_t$. Note that $U_t S_t V_t^\top$ is referred to as the "compact" or "Reduced" SVD of $B_t$. Muon implements this algorithm using Newton-Schulz matrix iterations to approximately compute $U_t V_t^\top$ instead of directly employing SVD to improve computational efficiency. The relationship between (11), with $B_t = G_t$, and (2) is interpreted in Bernstein and Newhouse [2024b] (see Story II). In the following discussion, we refer Muon as the one described in (11). We first show that ASGO and Muon, without gradient accumulation and momentum, are equivalent.

**Proposition 1.** *If in ASGO (Algorithm 2 version) we set $\beta_1 = \beta_2 = 0, \epsilon = 0$, and $\tau = 1$, and in Muon (11) we set $\mu = 0$, ASGO is equivalent to Muon.*

*Proof.* Let $U_t S_t V_t^\top$ be the compact SVD of $G_t$. For ASGO, the update is

$$W_{t+1} = W_t - \eta_t (G_t G_t^\top)^{-\frac{1}{2}} G_t$$

$$\begin{aligned}
&= W_t - \eta_t (U_t S_t S_t^\top U_t^\top)^{-\frac{1}{2}} U_t S_t V_t^\top \\
&= W_t - \eta_t U_t (S_t^2)^{-\frac{1}{2}} U_t^\top U_t S_t V_t^\top \\
&= W_t - \eta_t U_t S_t^{-1} S_t V_t^\top \\
&= W_t - \eta_t U_t V_t^\top .
\end{aligned}$$

Clearly, this is the same as Muon. $\square$

We now prove a nonconvex convergence result for Muon in the deterministic case with $\mu = 0$.

**Theorem 4** (Nonconvex convergence of Muon). *We consider Muon presented in* (11) *with $\mu = 0$. In the deterministic case, i.e. no gradient noise, if we assume that there exists a lower bound $f^*$ such that $f(W) \geq f^*$ for all $W$ and Assumption* 2, *by taking $\eta_t \equiv \eta = \sqrt{\frac{2(f(W_0) - f^*)}{\|L\|_* T}}$, it holds that*

$$\frac{1}{T} \sum_{t=0}^{T-1} \|\nabla f(W_t)\|_* \leq \sqrt{\frac{\|L\|_* \, (f(W_0) - f^*))}{2T}}.$$

*Proof.* Since we assume the deterministic setting, we have $G_t = \nabla f(W_t)$. Also, since we assume $\mu = 0$, we have $U_t, V_t$ are the left and right singular vectors of $G_t$ such that $G_t = U_t S_t V_t^\top$. Then based on the smoothness assumption, it holds that

$$\begin{aligned}
f(W_{t+1}) &\leq f(W_t) + \langle \nabla f(W_t), W_{t+1} - W_t \rangle + \frac{1}{2} \mathrm{tr}\left( (W_{t+1} - W_t)^\top L (W_{t+1} - W_t) \right) \\
&= f(W_t) - \eta_t \mathrm{tr}\left( G_t^\top U_t V_t^\top \right) + \frac{\eta_t^2}{2} \mathrm{tr}\left( (U_t V_t^\top)^\top L (U_t V_t^\top) \right) \\
&= f(W_t) - \eta_t \mathrm{tr}\left( V_t S_t U_t^\top U_t V_t^\top \right) + \frac{\eta_t^2}{2} \mathrm{tr}\left( V_t U_t^\top L U_t V_t^\top \right) \\
&= f(W_t) - \eta_t \mathrm{tr}\left( V_t S_t V_t^\top \right) + \frac{\eta_t^2}{2} \mathrm{tr}\left( U_t^\top L U_t V_t^\top V_t \right) \\
&= f(W_t) - \eta_t \mathrm{tr}\left( S_t V_t^\top V_t \right) + \frac{\eta_t^2}{2} \mathrm{tr}\left( L U_t U_t^\top \right) \\
&\leq f(W_t) - \eta_t \|G_t\|_* + \frac{\eta_t^2}{2} \|L\|_* ,
\end{aligned}$$

where the last inequality is because of the fact that $UU^\top \preceq I$ and Lemma 1. Then by summing up over $t$ and rearrangement, we can obtain that

$$\sum_{t=0}^{T-1} \eta_t \|G_t\| \leq f(W_0) - f(W_T) + \sum_{t=0}^{T-1} \frac{\eta_t^2}{2} \|L\|_* .$$

Then we take $\eta_t \equiv \eta$ to obtain that

$$\frac{1}{T} \sum_{t=0}^{T-1} \|\nabla f(W_t)\| \leq \frac{f(W_0) - f^*}{\eta T} + \frac{\eta}{2} \|L\|_* .$$

Setting $\eta = \sqrt{\frac{2(f(W_0) - f^*)}{\|L\|_* T}}$ finishes the proof. $\square$

# H   Incorporating Projection in ASGO

To achieve fully theoretically rigorous convergence bounds for ASGO, i.e., with $D_{\mathrm{op}}$ and $D_{\mathrm{F}}$ strict constants as $T$ increases, one well-established way is to incorporate a projection operation in the Algorithm [Hazan et al., 2007, Duchi et al., 2011]. Consider a closed convex set $\mathcal{W} \subseteq \mathbb{R}^{m \times n}$ with an optimum $W_*$ inside and diameters $\tilde{D}_{\mathrm{op}}, \tilde{D}_{\mathrm{F}}$ such that for any $W_1, W_2 \in \mathcal{W}$, it holds that

$$\|W_1 - W_2\|_{\mathrm{op}} \leq \tilde{D}_{\mathrm{op}} \quad \text{and} \quad \|W_1 - W_2\|_{\mathrm{F}} \leq \tilde{D}_{\mathrm{F}}.$$

For Algorithm 1, we desire to search solution inside $\mathcal{W}$ and change the update step line 8 to

$$Z_{t+1} = W_t - \eta_t \Lambda_t^{-1} G_t \quad \text{and} \quad W_{t+1} = \Pi_{\Lambda_t}^{\mathcal{W}}(Z_{t+1}),$$

where the anisotropic projection operator $\Pi_{\Lambda_t}^{\mathcal{W}} : \mathbb{R}^{m \times n} \to \mathcal{W}$, as also incorporated in Hazan et al. [2007], Duchi et al. [2011], is defined as

$$\Pi_{\Lambda_t}^{\mathcal{W}}(X) \triangleq \underset{Y \in \mathcal{W}}{\arg\min} \|X - Y\|_{\Lambda_t}^2$$

for any $X \in \mathbb{R}^{m \times n}$, which enjoys the non-expansiveness of a projection operator:

$$\|\Delta W_{t+1} - W_*\|_{\Lambda_t}^2 = \text{tr}\left(\Delta W_{t+1}^\top \Lambda_t \Delta W_{t+1}\right) \leq \text{tr}\left(\Delta Z_{t+1}^\top \Lambda_t \Delta Z_{t+1}\right) = \|\Delta Z_{t+1}\|_{\Lambda_t}^2 \quad (12)$$

with $\Delta W_{t+1} \triangleq W_{t+1} - W_*$ and $\Delta Z_{t+1} \triangleq Z_{t+1} - W_*$ since $W_* \in \mathcal{W}$. Substituting this non-expansiveness property in the proof of nonsmooth convergence, or more specifically, in (6), we can obtain that

$$2 \sum_{t=0}^{T-1} \mathbb{E}[f(W_t)] - f(W_*)$$

$$\leq \frac{1}{\eta} \mathbb{E}\left[\sum_{t=0}^{T-1} \text{tr}\left(\Delta W_t^\top \Lambda_t \Delta W_t - \Delta Z_{t+1}^\top \Lambda_t \Delta Z_{t+1}\right)\right] + \eta \mathbb{E}\left[\sum_{t=0}^{T-1} \text{tr}\left(G_t^\top \Lambda_t^{-1} G_t\right)\right]$$

$$\overset{(12)}{\leq} \frac{1}{\eta} \mathbb{E}\left[\sum_{t=0}^{T-1} \text{tr}\left(\Delta W_t^\top \Lambda_t \Delta W_t - \Delta W_{t+1}^\top \Lambda_t \Delta W_{t+1}\right)\right] + \eta \mathbb{E}\left[\sum_{t=0}^{T-1} \text{tr}\left(G_t^\top \Lambda_t^{-1} G_t\right)\right]$$

$$= \frac{1}{\eta} \mathbb{E}\left[\sum_{t=0}^{T-1} \text{tr}\left(\Delta W_t^\top \Lambda_t \Delta W_t - \Delta W_t^\top \Lambda_{t-1} \Delta W_t\right)\right] + \frac{1}{\eta} \text{tr}\left(\Delta W_0^\top \Lambda_{-1} \Delta W_0\right) - \frac{1}{\eta} \mathbb{E}\left[\text{tr}\left(\Delta W_T^\top \Lambda_{T-1} \Delta W_T\right)\right]$$

$$+ \eta \mathbb{E}\left[\sum_{t=0}^{T-1} \text{tr}\left(G_t^\top \Lambda_t^{-1} G_t\right)\right]$$

$$\leq \frac{1}{\eta} \text{tr}\left(\Delta W_0^\top \Lambda_{-1} \Delta W_0\right) + \frac{1}{\eta} \mathbb{E}\left[\sum_{t=0}^{T-1} \text{tr}\left(\Delta W_t^\top (\Lambda_t - \Lambda_{t-1}) \Delta W_t\right)\right] + \eta \mathbb{E}\left[\sum_{t=0}^{T-1} \text{tr}\left(G_t^\top \Lambda_t^{-1} G_t\right)\right],$$

which yields exactly the same result as (6). Since the rest of the proofs of both nonsmooth and smooth theory are based on (6), we can achieve the same convergence results as Theorems 1 and 3, while we have the upper bound on $\max_{0 \leq t \leq T-1} \|W_t - W_*\|_{\text{op}} \leq \tilde{D}_{\text{op}}$ and $\max_{0 \leq t \leq T-1} \|W_t - W_*\|_{\text{F}} \leq \tilde{D}_{\text{F}}$, where the diameters $\tilde{D}_{\text{op}}$ and $\tilde{D}_{\text{F}}$ depend on the geometry of the convex set $\mathcal{W}$ and remain constants when $T$ increases.

