# OpenReview forum: "ASGO: Adaptive Structured Gradient Optimization"
_NeurIPS.cc/2025/Conference — NeurIPS 2025 poster_

### Official Review · Reviewer_hvYf · 2025-07-01

**Clarity:** 3
**Significance:** 2
**Originality:** 3
**Rating:** 4
**Confidence:** 3

**Summary:**

The authors propose ASGO (Adaptive Structured Gradient Optimization), a new structured gradient method that exploits the matrix structure of gradients, including their low-rank and block-wise diagonal properties. ASGO improves the convergence guarantees of the existing algorithm Shampoo and is compared with the existing algorithm Muon. Experiments on language model tasks show that ASGO performs competitively, matching or outperforming methods such as AdaGrad, Adam, Muon, and Shampoo.

**Questions:**

1.line 113, in the update equation: would it be clearer to use  $\Lambda_t$  instead of $V_t$, since Algorithm 1 (line 8) uses $\Lambda_t$, which is a slight modification of $V_t$?
2.line 239: you claim an equivalence between ASGO and Muon (without momentum) and draw an analogy to AdaGrad and SignSGD. Could the authors provide a more concrete explanation for this equivalence?
3.line 258, Remark 4: In the smooth setting, how the convergence result of Muon compared with your result in Theorem 3?

**Ethical Concerns:**

["NO or VERY MINOR ethics concerns only"]

**Final Justification:**

Thanks for the authors' responses. According to other reviewers' comments and the authors' responses, I still remain the original score for this paper.

**Limitations:**

yes

**Quality:**

3

**Strengths And Weaknesses:**

Strengths:
The paper presents clear and well-structured discussions. The convergence analysis and experimental evaluations are thorough and well-executed, particularly the comparisons with existing methods.

Weaknesses:
1.Although ASGO shows comparable performance to Muon and other baselines, the improvements in final accuracy or perplexity are not consistently significant.
2.The discussion of nonconvex convergence is mostly conjectural, with key claims for ASGO inferred from Muon’s behavior rather than being directly proven.

---

> ### Author Rebuttal · Authors · 2025-07-31
>
> We sincerely appreciate the reviewer's recognition of our paper. Many thanks! Below, we address the weakness and question parts of the review.
>
> **Weakness 1: The experimental results are not consistently significant.**
>
> We thank the reviewer for their insightful comment. You are right to point out the significance of the empirical results.
> The central contribution of this paper is to establish the theoretical motivation and empirical viability of structure-aware preconditioning. Specifically, we want to demonstrate why and how a structure-aware optimizer can outperform a structure-agnostic one like Adam. Based on this motivation, ASGO is designed to overcome this by employing preconditioners that explicitly model the rich interactions within a parameter tensor (e.g., the row-and-column structure of a matrix), which a structure-agnostic optimizer (like AdamW) cannot.
>
> To provide stronger empirical validation for this core thesis, we have conducted a new, more rigorous pre-training experiment for a 125M parameter GPT-2 model on OpenWebText, increasing the token budget to 1.26 billion tokens more than double our original run. This budget represents approximately 10 tokens per parameter, which attains the limit of our computational resources. We adopted the Newton-Schultz(NS) algorithm [1, Thm. 5.2] for 10 steps instead of Singular Value Decomposition (SVD) to compute the inverse square root of preconditioning. which substantially reduces the wall-clock time of the optimizer step.
>
> | **Optimizer** | **315M** | **630M** | **945M** | **1.26B** |
> | :--- | :---: | :---: | :---: | :---: |
> | AdamW | 4.71 | 4.05 | 3.79 | 3.73 |
> | Muon | 4.17 | 3.77 | 3.62 | 3.58 |
> | **ASGO (Ours)** | 4.11 | 3.74 | 3.58 | 3.53 |
>
> Table 1: Training loss at different checkpoints for the 125M GPT-2 model on OpenWebText. The results are from the extended 1.26B token training run.
>
> After a fair hyperparameter search, the results (Table 1) show that structure-aware optimizers (our ASGO and Muon) achieve a consistently lower final training loss than the structure-agnostic AdamW. This new result directly validates our paper's central argument: that leveraging parameter structure leads to a better-trained neural network.
>
> Furthermore, we analyzed the practical wall-clock time by using the efficient Newton-Schultz (NS) method (10 steps) to compute the preconditioner's inverse square root. While the optimizer step for ASGO is computationally more intensive than AdamW, the total training time is remarkably similar across all methods, as shown in Table 2. This demonstrates that the per-step overhead of ASGO is not a bottleneck for overall training throughput.
>
> | **Metric** | **Adam** | **Muon** | **ASGO (Ours)** |
> | :-------------- | :----------: | :----------: | :-------------: |
> | WCT             | 127.76 min   | 129.01 min   | 133.50 min      |
> | Train Loss      | 3.73         | 3.58         | 3.53            |
>
> Table 2: The total Wall-clock Time(WCT) and final train loss for training 1200 steps
>
> We believe that by leading with our theoretical motivation and supporting it with these new, stronger empirical results, we have addressed the reviewer's concern. We will update the manuscript with this new analysis. We plan to conduct more experiments
> and include these in future revisions.
>
> **Weakness 2: The discussion of nonconvex convergence is mostly conjectural.**
>
> The nonconvex convergence of ASGO is indeed conjectured from Muon's convergence guarantee, based on the close relation between the algorithms. We think it is an interesting future research direction, and we will actively work on this. Thanks a lot for raising this point!
>
> **Question 1: Use $\Lambda_t$ instead of $V_t$ in line 113.**
>
> We thank the reviewer for the careful reading and the helpful suggestion. We will modify the formulation in line 113 and add a more detailed definition in future revisions accordingly.
>
> **Question 2: More discussions on the relation between Muon and ASGO.**
>
>
> Thank you for this excellent question, which allows us to elaborate on a key intuition behind our work. First, we want to give some intuition on the relationship between Muon and SignSGD. As analyzed in [2], both Muon (without momentum) and SignSGD (without momentum) can be interpreted in the steepest descent framework (Equation (2) in the paper), but with respect to different matrix norms. This provides a clear theoretical connection. Meanwhile, both methods can be seen as performing whitening on the first-order momentum. SignSGD performs element-wise whitening while Muon performs spectral whitening.
>
> Then we kindly lead you to the connection between ASGO and Muon. In Section 6, we note that ASGO without accumulating $GG^\top$ in the preconditioner is equivalent to Muon without momentum, and provide a corresponding proof in Appendix G. To be more concrete, this equivalence comes from the equation $(G_tG_t^\top)^{-1/2} G_t = U_t V_t^\top$, where $U_t, V_t$ are the left and right singular vectors in the compact singular value decomposition of $G_t$, i.e., $G_t = U_t \Sigma_t V_t^\top$. Intuitively, ASGO uses a smoother preconditioning than Muon, since it updates by $(\sum_{s=1}^{t} G_sG_s^\top)^{-1/2} G_t$ instead of directly using $(G_tG_t^\top)^{-1/2}$ as the preconditioner.
> This is similar to the %relation
> relationship
> between AdaGrad and SignSGD, since if we remove the accumulation in the preconditioner of AdaGrad and the momentum of SignSGD, we can also obtain equivalence between the two algorithms:
> $$
> \text{AdaGrad:} \quad w_{t+1} = w_{t} - \eta_t (g_t^2)^{-1/2} g_t
> $$
> $$
> \text{SignSGD:} \quad w_{t+1} = w_t - \eta_t \text{sign}(g_t)
> $$
> where $(g_t^2)^{-1/2}$ here denotes coordinate-wise operations. Based on this similarity between the relations, we say the connection between ASGO and Muon is analogous to the connection between AdaGrad and SignSGD.
> Thanks for this insightful question, and we will also add the discussion context to the paper to make it clearer.
>
> **Question 3: How does the convergence of Muon compare with Theorem 3.**
>
> Thanks for this important question. Though the results are both obtained under smooth settings, we kindly note that the two results are generally incomparable, since Theorem 3 indicates the loss convergence of ASGO under convex settings, while Equation (3) introduces the convergence rate of Muon to a stationary point under nonconvex settings. We consider the nonconvex convergence of ASGO as an interesting topic for future work, and expect that it should be on par with the convergence rate in Equation (3) for Muon.
>
> **References**
>
> [1] Nicholas J. Higham. Functions of Matrices: Theory and Computation.
>
> [2] Jeremy Bernstein, Laker Newhouse. Old Optimizer, New Norm: An Anthology

---

### Official Review · Reviewer_uDaJ · 2025-07-02

**Clarity:** 1
**Significance:** 2
**Originality:** 2
**Rating:** 2
**Confidence:** 3

**Summary:**

This paper proposes ASGO (Adaptive Structured Gradient Optimization), an optimization algorithm that exploits structured properties of neural network gradients by using a single preconditioner matrix, requiring less memory than existing structured methods like Shampoo. The authors provide theoretical analysis and demonstrate empirical performance on language modeling tasks.

**Questions:**

- How does the computational cost of computing $V_t^{-1}$ scale with dimension m?
- How does this compare to the computational cost of AdamW or Muon in practice on scale?

The paper covers an interesting problem but it contains multiple false claims and provides a weak experimental evaluation. In my opinion, an accept would require the correction of all weaknesses I listed above and an evaluation either on a broader benchmark or on larger language modes (or a strongly limited scope of the paper but this reduces the benefit). Also it should include a runtime analysis and a comparison with other optimizers given the same compute budget (for the runtime but also for the HPO).

**Ethical Concerns:**

["NO or VERY MINOR ethics concerns only"]

**Final Justification:**

The authors addressed many of my points in the review but one major criticism is unsolved. Either one could introduce a general optimizer but have to evaluate it on common benchmarks like AlgoPerf or DeepOBS, or one introduces a specialized optimizer for small language models but communicate this limitation clearly. But introducing a general optimizer and viewing the proper evaluation as future work doesn't work in my opinion. So far, we only know that the general optimizer works on 125M LLM sub-epoch training.
Also, the lack of source code to test the performance and runtime is in this case not acceptable. Therefore, I keep my score.

**Limitations:**

No, the paper does not cover the cost of the matrix inversion and how it behaves on larger models.

**Paper Formatting Concerns:**

No.

**Quality:**

1

**Strengths And Weaknesses:**

### Strengths

- ASGO requires only one preconditioner matrix compared to Shampoo's two, reducing memory requirements by approximately half while achieving better theoretical convergence rates.
- The paper clearly explains why single-side preconditioning may be preferable.


### Weaknesses

- The authors claim that "adaptive gradient methods like AdaGrad and Adam" “apply a diagonal matrix preconditioner to the gradient" (line 19). This is wrong, also Adam does not use matrix inversion $Λt^(-1)$ - it uses element-wise division by the square root of the second moment estimates. This is a fundamental misrepresentation of how Adam actually operates.
- The authors state "adaptive gradient methods like Adam, treat the parameters as vectors and ignore the matrix structure of gradients." (line 30) This is incorrect. Adam performs element-wise operations on parameters regardless of their shape (vector, matrix, or tensor) - it doesn't reshape matrices into vectors.
- The authors introduce a new optimizer for deep neural networks, “like Adam”, but evaluate only on small language model tasks. In my opinion one could introduce a general optimizer but have to evaluate it on common benchmarks like AlgoPerf or DeepOBS, or one introduces a specialized optimizer for small language models but communicate this limitation clearly.
- In section 3.2, where the algorithm is introduced, it is unclear what $V_t$ is.
- The claim “DNNs are generally nonconvex globally, convexity may apply locally in some regions” (line 121) is not backed.
- In line 236, “Moun computes ….”, I can’t see how this represents the Muon optimizer.
- The paper refers to Algorithm 2 and to Algorithm 3 (line 265, 267, 271) which are not in the paper.
-  The authors perform in 7.2 a training on a ~125M parameter model on a token budget of 1000 steps x 32bs x 8 grad accum x 4 GPUS x 512 seq length = 525M tokens which is way below Chinchilla optimal. If one proposes an optimizer for language model pretraining, I would like to see experiments on at least 1B and on at least 8x Chinchilla optimal token budgets (at least for smaller models).
- The optimizer contains a matrix inversion, this is any time consuming operation, however the authors do not discuss or provide measurements on the runtime of the optimizer. A fair comparison would include a comparison on the same runtime (compute budget) against other optimizers.
- In contrast to the "YES" in the Paper Checklist 5, the authors do not provide code to replicate the experiments or test the optimizer. This is crucial for a newly introduced optimizer.

---

> ### Author Rebuttal · Authors · 2025-07-31
>
> We thank the reviewer for the constructive and detailed feedback.
>
> **On Weaknesses 1 \& 2: Diagonal preconditioner**
>
> We agree that Adam and AdaGrad are implemented using element-wise operations. However, our claim is that this element-wise division is mathematically equivalent to preconditioning with a diagonal matrix. This is a standard and widely accepted viewpoint in the optimization literature, and Adam is described in this fashion.
> Let's consider the update rule for Adam:
> $$
> \theta_{t+1}=\theta_t-\eta \cdot m_t /\left(\sqrt{v_t}+\epsilon\right),
> $$
> where $m_t$ ($v_t$) is the first-order (second-order) momentum and the division is elementwise. This operation is identical to:
> $$
> \theta_{t+1}=\theta_t-\eta \cdot P_t^{-1} m_t
> $$
> where $P_t = \operatorname{diag}\left(\sqrt{v_t}+\epsilon\right)$ is a diagonal preconditioner. The inverse of this diagonal matrix, $P_t^{-1}$, is simply a diagonal matrix with elements $1 /\left(\sqrt{v_{t, i}}+\epsilon\right)$, and
> pre-multiplying the momentum
> gradient $m_t$ results in the element-wise division. This interpretation is discussed in foundational texts, e.g., [1, Chapter 5.4]. The original AdaGrad paper also introduced both full-matrix and diagonal versions [2].
>
> This directly relates to the reviewer's second point. Since Adam performs element-wise operations, its preconditioning model is conceptually equivalent to flattening all parameters (e.g., matrices $W \in \mathbb{R}^{d_1 \times d_2}$ ) into a single long vector and then applying the diagonal preconditioner discussed above. This approach implies that each parameter is scaled independently, a decoupling that renders the preconditioner agnostic to the parameters' inherent tensor structure.
> This is why we say Adam generally ignores the structure of parameters.
>
> This exact limitation of Adam is the primary motivation for our work. ASGO is designed to employ preconditioners that explicitly model the rich interactions within a parameter tensor (e.g., the row-and-column structure of a matrix), which a diagonal preconditioned method like Adam cannot.
>
> **On Weakness 3: Evaluation on benchmarks**
>
> The primary objective of our experiments was to provide a clear and controlled empirical validation of our theoretical analysis, specifically demonstrating why and how a structure-aware optimizer can outperform a structure-agnostic one like Adam. For this purpose, small- to medium-scale language models provide an ideal and interpretable testbed.
> This focused approach also aligns with our available computational resources, allowing us to conduct a thorough and rigorous investigation. Our experimental setting is similar in spirit to that of other recently proposed optimizers; for instance, Muon [3] was also initially evaluated on similarly-scaled models (e.g., NanoGPT) to establish its fundamental properties.
>
> Furthermore, to demonstrate the versatility of ASGO, we extended our evaluation beyond pre-training to include fine-tuning experiments, showcasing its potential in both training from scratch and adaptation scenarios.
>
> While the central contribution of this paper is to establish the theoretical motivation and empirical viability of structure-aware preconditioning, we concur with the reviewer that broader evaluation is a crucial next step. We appreciate the suggestions of benchmarks like AlgoPerf and DeepOBS, and we will prioritize them in future work to rigorously assess the scalability and generalizability of ASGO.
>
> **On weakness 4: Definition of $V_t$**
>
> $V_t$ is defined in Line 6 of Algorithm 1, (cited on the first line of Sect. 3.2, but which the Latex editor put on the next page in Sect. 4). We will add the definition where $V_t$ first appears.
>
> **On weakness 5: Local convexity**
>
> Our claim is based on the widely studied property that while DNN loss landscapes are globally non-convex, they often possess favorable local geometric properties. This is supported by a growing body of theoretical work. For instance, [6] introduces a class of functions satisfying local strong convexity [6, Assumption 1] and shows that deep ReLU networks are a practical realization of this class [6, Section 4.2]. Similarly, [7] establishes the restricted strong convexity of DNNs within a sequence of ball-shaped regions around the weight iterates. In our revision, we will add a discussion of this context and include these supporting citations.
>
> **On weakness 6: Muon updates**
>
> The statement on Line 236 and the accompanying Equation (2) describe the momentum-free update step of the Muon optimizer. This step corresponds to a steepest descent update under the spectral norm, a definition that is detailed in [4, Proposition 5]. In the revised manuscript, we will restructure this section to make the statement unambiguous.
>
> **On weakness 7: Algorithm 2 and 3**
>
> Algorithm 2 is in Appendix B page 15, Algorithm 3 is in Appendix C page 17. In the revised manuscript, we will ensure that every reference to these algorithms is accompanied by an explicit note directing the reader to the correct appendix.
>
> **On weakness 8: Larger training size**
>
> We agree that evaluating optimizers on a larger, more representative token budget is essential for pre-training language models. Consequently, we have conducted a new pre-training experiment for the 125M parameter GPT-2 model on the OpenWebText dataset with a significantly larger token budget.
>
> We trained the model for 1200 steps using a batch size of 64, 4 A6000 GPUs, a sequence length of 512, and 8 gradient accumulation steps, yielding a total of 1.26B tokens. This budget represents approximately 10 tokens per parameter, which attains the limit of our computational resources.
>
> Although we list a fair Hyperparameter Parameter Optimization (HPO) in Appendix C.2, we performed a new HPO for the learning rate of all three optimizers under this new setting:
>
> AdamW: 1r in $\{5 e^{-4},1 e^{-3},5 e^{-3},1 e^{-2}\}$
>
> Muon \& ASGO: 1r in $\{1 e^{-3},5 e^{-3},1 e^{-2},5 e^{-2}, 1 e^{-1}\}$
>
> | Optimizer | **315M** | **630M** | **945M** | **1.26B** |
> | :--- | :---: | :---: | :---: | :---: |
> | AdamW | 4.71 | 4.05 | 3.79 | 3.73 |
> | Muon | 4.17 | 3.77 | 3.62 | 3.58 |
> | **ASGO** | 4.11 | 3.74 | 3.58 | 3.53 |
>
> Table 1: Training loss at different checkpoints for the 125M GPT-2 model on OpenWebText.
>
> As Table 1 clearly indicates, the structure-aware preconditioning methods (Muon and our ASGO) consistently achieve a lower final training loss compared to the vector-based AdamW optimizer when trained on a larger token budget. This new experiment provides strong empirical evidence for our paper's central thesis: that leveraging the matrix structure of parameters is beneficial for training DNNs. We will replace our previous results with this new, more rigorous experiment in the revised manuscript. We plan to conduct more experiments and include these in future revisions.
>
> **On weakness 9: Runtime comparison**
>
> First, our practical implementation for all experiments follows Algorithm 2, which directly computes $\left(V_t+\epsilon I_n\right)^{-1 / 2}$ instead of the matrix inverse. This is mathematically identical to the formulation in Algorithm 1.
>
> Second, our initial version of ASGO relied on SVD to compute $\left(V_t+\epsilon I_n\right)^{-1 / 2}$. This approach carries a significant computational burden. For the new experiments presented in this rebuttal, we have adopted the Newton-Schultz (NS) iterative method, which substantially reduces the wall-clock time (WCT) of the optimizer step and makes the algorithm more practical. The following runtime analysis is based on this improved NS-based implementation.
>
> As shown in [5, Thm. 5.2], computing $A^{-1 / 2}$ can be achieved by computing the matrix of a block matrix
> $$
> \operatorname{sign}\left(\left[\begin{array}{cc}
> 0 & A \\\\
> I & 0
> \end{array}\right]\right)=\left[\begin{array}{cc}
> 0 & A^{1 / 2} \\\\
> A^{-1 / 2} & 0
> \end{array}\right],
> $$
> which is the same operation as computed in Muon. This implies that computing the square root inverse in ASGO shares the same order of complexity as computing the matrix sign function in Muon.
>
> To provide a concrete analysis, we measured the wall-clock time under the exact settings of our 1.26 B token experiment.
> | **Optimizer** | **Single Train Step WCT** |
> | :--- | :---: |
> | AdamW | 5.451 sec |
> | Muon | 5.519 sec |
> | **ASGO (Ours)** | 5.947 sec |
>
> Table 2: Wall-clock time per training step.
>
> While ASGO's optimizer step is more expensive than AdamW, the forward/backward pass is the dominant bottleneck. The table 3 shows total end-to-end training time is only marginally affected.
>
> | **Metric** | **AdamW** | **Muon** | **ASGO (Ours)** |
> | :--- | :---: | :---: | :---: |
> | WCT | 127.76 min | 129.01 min | 133.50 min |
> | Train Loss | 3.73 | 3.58 | 3.53 |
>
> Table 3: The total Wall-clock Time (WCT) and final training loss for training 1200 steps.
>
> We think this additional experiment also addresses the two questions of the reviewer.
>
> **On weakness 10: Public code**
>
> We apologize for the confusion caused by the checklist. As stated in our justification for Checklist 5, we will make the complete source code for our experiments publicly available in a GitHub repository upon acceptance of the paper.
>
> We believe that we have addressed the weaknesses and questions mentioned in the review, and hope that it convinces the reviewer to change the negative view of our paper.
>
> **References**
>
> [1] R.-y. Sun. Optimization for Deep Learning: An Overview
>
> [2] J. Duchi et al. Adaptive Subgradient Methods for Online Learning and Stochastic Optimization
>
> [3] K. Jordan et al. Muon: An optimizer for hidden layers in neural networks
>
> [4] J. Bernstein te al. Old Optimizer, New Norm: An Anthology
>
> [5] N. J. Higham. Functions of Matrices: Theory and Computation
>
> [6] F. Liao et al. Provable Accelerated Convergence of Nesterov’s Momentum for Deep ReLU Neural Networks
>
> [7] A. Banerjee et al. Restricted strong convexity of deep learning models with smooth activations

---

### Official Review · Reviewer_WuZT · 2025-07-03

**Clarity:** 3
**Significance:** 3
**Originality:** 2
**Rating:** 5
**Confidence:** 4

**Summary:**

In light of the recent resurgence of "matrix" layer-wise preconditioners in deep learning optimization such as Shampoo and Muon, this paper considers understanding the principled benefit of these methods via regret / convergence bounds. In particular, the authors present an algorithm denoted Adaptive Structured Gradient Optimization (ASGO); compared to Shampoo and Muon which can be viewed as preconditioning the gradient matrix in weight space from both sides, ASGO proposes only storing a one-sided preconditioner. Though seemingly a more restrictive optimizer class, the authors show that from the view of convex or smooth matrix-variable optimization, the convergence upper bound of ASGO may exhibit more favorable dependence on problem parameters than Shampoo or Muon, or even full-matrix Adagrad. These hints from theory are corroborated in experiments, where ASGO is demonstrated to consistently match the performance of other state-of-the-art methods on various deep learning tasks, indicating that it may suffice to perform salient one-sided preconditioning of the gradient in practice.

**Questions:**

Besides the points laid out under Weaknesses, I have the following auxiliary questions:

1. A critical aspect of deep learning optimization is ensuring "scale-invariance" of the hidden activations across width/depth, such as ensured by $\mu$P [1,2] or spectral scaling [3] and developments thereafter. How should ASGO be scaled layer-wise to ensure stability of hyperparameters across architecture sizes?

2. Regarding ASGO's compatibility with $d > 2$ dimensional tensor weights, what prevents handling higher-dimensional tensors as in, for example, Muon?

[1] Yang and Hu, "Feature Learning in Infinite-Width Neural Networks"

[2] Yang et al, "Tensor Programs V: Tuning Large Neural Networks via Zero-Shot Hyperparameter Transfer"

[3] Yang et al, "A Spectral Condition for Feature Learning"

**Ethical Concerns:**

["NO or VERY MINOR ethics concerns only"]

**Final Justification:**

This paper presents an interesting new member of the "Factored Adagrad" family. Interestingly, it is argued that a "weaker" class of preconditioners --- in this case one-sided preconditioning versus, e.g., Shampoo's two-sided one --- can actually lead to performance **improvement**. Some theory via optimization over a matrix-shaped variable is derived to justify this phenomenon by demonstrating different preconditioning schemes lead to different manifestations of problem-parameters, where certain preconditioners can outperform other "stronger" ones if these problem-parameters line up in the right way.

The theory admittedly does not capture a full story in relation to deep learning, e.g. restricted to convex optimization over a single-block matrix variable, comparisons of performance **upper bounds** etc, which most deep learning-oriented methods tend to lack. However, more importantly, the authors support the efficacy of their optimizer with some LLM pretraining experiments, showing performance on-par with the state-of-the-art in Muon, with similarly general algorithmic set-up. The main difference is that ASGO maintains an Adagrad-like moment accumulator/preconditioner, while Muon performs orthonormalization per-step, which gives Muon the edge in computational efficiency via more specific iterative methods tailored to orthonormalization rather than generic matrix inverse sqrts. However, as understood by the deep learning optimization community, for large networks such as in LLMs, the FLOPs introduced by layer-wise preconditioning (within reason) is quite small proportional to a backwards pass.

Putting these contributions together, I think various communities will find use in these results, and recommend acceptance (5).

**Limitations:**

Yes

**Paper Formatting Concerns:**

None.

**Quality:**

3

**Strengths And Weaknesses:**

Overall, this paper presents an interesting perspective to gradient preconditioning for deep learning optimization. In particular, the observation that a "weaker" preconditioner class might actually yield better results is nominally surprising and potentially interesting for the community. Furthermore, strong empirical results and implementation details are provided, which will be helpful for the community. Therefore, I recommend acceptance of the paper.

**Strengths**

To motivate where a smaller preconditioner can actually benefit convergence, convergence rate upper bounds are provided for convex non-smooth and smooth settings where the optimization variable is matrix-shaped, demonstrating the performance bounds on ASGO can exhibit better dependence on certain instance-dependent quantities than Shampoo or full-matrix Adagrad. This serves as evidence that there may not be a monotonic relationship between the memory/algorithmic complexity of a preconditioner and practical performance.

To complement the theoretical observations, the paper provides solid experiments on standard GPT2 pretraining and finetuning benchmarks, showing that its performance can outperform standard AdamW baselines, while nearly matching Muon. Empirically, it is also noted that ASGO may be generally more robust to learning rate selection than Muon.

**Weaknesses**

My main concern regarding the theory of in the paper are as follows:

1: Conclusions are drawn by comparing performance upper bounds. For the purpose of motivating empirical performance this may be fine, but a rigorous comparison should provide performance lower bounds on, e.g. full-matrix Adagrad on classes of instances, where it may then be shown that ASGO's upper bounds rigorously outpace other methods. In general, walking through some problem instances (even if simple) can greatly improve the claim that a simpler preconditioner can rigorously outperform larger ones.

2: The theoretical set-up considers a single matrix optimization variable. However, the non-triviality of establishing feature learning in neural network theory comes from the compositionality of layers. A simple example could be linear regression with two-layer linear networks $f(x) = W_2 W_1 x$. I am not confident the analysis in this paper can predict or describe the unique optimization geometry arising from matrix-valued weights interacting via composition. This also now adds the dimension of comparing **layer-wise** versus potentially cross-layer preconditioning (see e.g. KFAC variants [1] and PSGD [2]).

**Minor Comments**

Regarding the typical rank-structure / evolution in neural network weights, this paper may be insightful for the authors [3].

The statement of the Muon bound (3) seems wrong and differs from the more reasonable statement in Appendix G.


[1] Martens and Grosse, "Optimizing Neural Networks with Kronecker-factored Approximate Curvature"

[2] Li, "Preconditioned Stochastic Gradient Descent"

[3] Martin and Mahoney, "Implicit Self-Regularization in Deep Neural Networks: Evidence from Random Matrix Theory and Implications for Learning"

---

> ### Author Rebuttal · Authors · 2025-07-31
>
> We would like to extend our sincere appreciation to the reviewer for all the constructive questions and suggestions and recognition of our paper. The reviewer's remarks provide many important insights for future research directions relevant to ASGO. Many thanks!
>
> **Weakness 1: Conclusions are drawn by comparing performance upper bounds.**
>
>
> This is an important question. We agree with the reviewer that comparing upper bounds is not a rigorous way to compare the algorithms, and here we compare the upper bounds mainly to give some intuition on how ASGO can be potentially beneficial from a convergence guarantee viewpoint.
> We are definitely interested in finding lower bounds for the class of full-matrix AdaGrad or Shampoo, and some important scenarios where one-sided preconditioners can largely outperform more complex ones.
> Hence, this
> is an interesting future topic and may inspire even more ways to improve ASGO.
>
> **Weakness 2: The theoretical set-up considers a single matrix optimization variable.**
>
> Thanks a lot for this constructive and meaningful comment! Considering the layer-wise structure of the problem is definitely a very interesting topic. It is indeed the case that the analysis in the paper may not be trivially generalized to the setting where a lot of weight matrices are mixed together, as in neural networks. We consider this a potential future research topic and feel excited to work more on this question. For now, we think that the
> recent paper [1] may provide some viewpoints and intuitions into how to prove the properties with a layer-wise structure, where the authors establish the convergence rates under multi-layer settings. However, [1] still doesn't take the interactions between matrices into consideration, which can make the problem much more complex. It would be really interesting to further explore this.
>
>
> **Minor comments 1: The paper [2] for rank-structure / evolution in neural network weights.**
>
> We thank the reviewer for bringing this interesting and relevant paper to our attention.
> The structured properties of matrices in DNNs discussed in the paper, especially the weight rank dynamics during the training process, are really insightful. This provides some significant evidence on what the weight rank should be in the training process, which is also a critical component of the ASGO convergence rate, as presented in Theorems 1 and 3. We will incorporate a discussion on this point in future revisions. Thanks again for introducing this insightful paper!
>
>
>
> **Minor comments 2: The Muon bound (3) differs from the statement in Appendix G**
>
> Thanks for the careful reading and the question. We kindly note that we use different forms of the convergence rate in Equation (3) and Theorem 4 in Appendix G, and we can derive (3) based on Theorem 4:
> $$
> \frac{1}{T} \sum_{t=0}^{T-1} || \nabla f(W_t) || \leq \sqrt{\frac{|| L || (f(W_0) - f^*)}{2T}} \quad \Rightarrow \quad \min_{0\leq t \leq T-1} || \nabla f(W_t) ||^2 \leq || L || (f(W_0) - f^{\*}) / (2T).
> $$
> Here, the norms are the trace norm $ || \cdot ||_{\*} $.
> Therefore, Equation (3) and Theorem 4 actually indicate the same convergence rate. We apologize for causing confusion because of the use of different forms and will make them consistent in future revisions.
>
> **Question 1: The use of $\mu$P.**
>
> This is definitely an important and interesting point for the scaling of ASGO. Intuitively, it is natural for ASGO to incorporate $\mu$P based on [3], just as it is for Muon. This is because we can expect the spectral norm of the weight update to have an approximately $ \mathcal{O}(1) $ scale (for the practical version (Algorithm 2)):
>
> $$
> || \Delta W || = || \Lambda_t^{-1} M_t || = || (\sum_{s=0}^{t} \beta^{t-s} (1-\beta) G_t G_t^\top)^{-1} M_t || \approx \mathcal{O}( || (G_t G_t^\top)^{-\frac{1}{2}} G_t || ) = \mathcal{O}(1).
> $$
> Here the norms are the spectral norm $ ||\cdot||_{op} $.
> Therefore, to implement $\mu$P for ASGO, we only need to use an adjusted learning rate $\tilde{\eta}_t = \sqrt{n/m} \eta_t $ to fulfill the requirement $|| \Delta W || = \mathcal{O}(\sqrt{n/m})$, and set the initialization accordingly. This formulation is straightforward and can be important for scaling up ASGO, which we consider an important future research direction.
>
> **Question 2: ASGO for higher-dimensional tensors.**
>
> We thank the reviewer for this insightful question. ASGO can indeed handle higher-dimensional tensors just like Muon, which basically flattens the tensor into 2-dimensional matrices. We are also considering other possible solutions that may utilize the structure of higher-dimensional tensors better, which could be an interesting future
> research topic.
>
>
> **References**
>
> [1] Riabinin, Artem, et al. "Gluon: Making Muon \& Scion Great Again! (Bridging Theory and Practice of LMO-based Optimizers for LLMs)." arXiv preprint arXiv:2505.13416 (2025).
>
> [2] Charles H. Martin and Michael W. Mahoney. "Implicit self-regularization in deep neural networks: Evidence from random matrix theory and implications for learning." Journal of Machine Learning Research 22.165 (2021): 1-73.
>
> [3] Greg Yang, James B. Simon, and Jeremy Bernstein. "A spectral condition for feature learning." arXiv preprint arXiv:2310.17813 (2023).

---

> ### Comment · Reviewer_WuZT · 2025-08-05
>
> I thank the authors for their clarifications. I maintain my positive evaluation.

---

> > ### Author Response · Authors · 2025-08-09
> > **Reply to Reviewer WuZT, Thanks!**
> >
> > We would like to provide our sincere appreciation again for all the interesting and constructive points and your recognition of our paper. We will be interested in exploring these directions in future work. Thanks for your dedication!

---

### Official Review · Reviewer_Vxtn · 2025-07-03

**Clarity:** 3
**Significance:** 3
**Originality:** 3
**Rating:** 4
**Confidence:** 3

**Summary:**

The paper designed an optimization algorithm when the decision variable has a matrix structure. The algorithm has some connections with AdaGrad. Compared with Shampoo, it does not need to maintain two preconditioning matrices. Theoretically, the authors provided the convergence guarantees for the convex nonsmooth setting and the convex smooth setting.

**Questions:**

### Major Question:

1. How should $\epsilon$ be chosen? Is it introduced to ensure the invertibility of $\Lambda$?

2. On Page 6, it is mentioned that the preconditioner should be on the right side of $G_t$. Why is it placed on the left side in Algorithm 1?

3. What does the sentence "we expect that ASGO has a nonconvex convergence rate comparable to (3) obtained by Muon up to logarithmic factors" mean? Are the authors aiming to establish a similar rate in the nonconvex regime?

4. Is there any comparison between ASGO and other algorithms in terms of wall-clock time?


### Minor Suggestions:

1. At the top of Page 5, I suggest including $T$ in the convergence rates for clarity.

2. When describing the algorithm in Section 3.2, I recommend defining $V$.

**Ethical Concerns:**

["NO or VERY MINOR ethics concerns only"]

**Final Justification:**

I raised my score because the authors provided a detailed explanation during the rebuttal on how to incorporate projections, which enhances the soundness of the theory.

**Quality:**

3

**Strengths And Weaknesses:**

### Strengths:

1. The authors provide a convergence guarantee for the algorithm. Furthermore, they characterize scenarios where the algorithm benefits, such as when the gradient is of low rank or when there is a block-wise diagonal Hessian structure. The notation introduced in Assumption 2 aids in this characterization.

2. The experimental results appear promising.



### Weaknesses:

1. The bounds in Theorems 1 and 3 include a term $D_{op} = \max_t ||W_t - W_*||$, which may grow with $T$. Consequently, it is not entirely rigorous to claim convergence, particularly in the absence of a bounded domain.

2. The paper compares ASGO with SGD. However, I am uncertain whether this is a fair comparison, as SGD can converge without requiring a bounded domain, and its complexity can depend only on $||W_0 - W_*||^F$.  This makes it challenging to directly compare with $D_{op}$.

Reference.
Orabona, Francesco. "Last iterate of SGD converges (even in unbounded domains)." Blogpost on http://parameterfree. com, available at https://parameterfree. com/2020/08/07/last-iterate-of-sgd-converges-even-in-unbounded-domains (2020).

---

> ### Author Rebuttal · Authors · 2025-07-31
>
> Thank you very much for your careful reading and insightful opinions and questions, which we now address.
>
> **Weakness 1: $D_{op}$ may grow with $T$.**
>
> Thanks for the meaningful question. It is indeed true that $D_{op}$ could increase when $T$ increases in the settings of Theorems 1 and 3, but as we also mentioned in Remark 1 (after Corollary 2), we can address this by invoking a projection onto a bounded convex set, where the projection should be with respect to $|| \cdot ||_{\Lambda_t}$. This technique is well-established in standard optimization literature [1,2]. Here, since the projection operation is rarely used in practical tasks like training DNNs, we follow [3] and omit it in the algorithm and theorem. See the discussion after Theorem 7 in [3] for a similar opinion.
>
> **Weakness 2: The comparison between ASGO and SGD may be unfair**
>
> 1. We agree with the reviewer that the comparison is not fully rigorous, but we included it to provide some intuition about when ASGO may have a good convergence guarantee and how ASGO can utilize structural properties (low-rank gradients for the nonsmooth part), and we are not claiming that ASGO is a better algorithm than SGD. Note that a similar comparison was also considered in, for example, the original AdaGrad paper [1], also for explaining the potential benefits of AdaGrad.
>
> 2. Also, it is possible to prove rigorously the convergence of ASGO in an unbounded domain, following [4] for AdaGrad. However, since this analysis introduces novel assumptions and can be much more complex, it goes beyond the scope of this paper, and we leave it as an interesting future direction of research.
>
> **Major question 1: Choosing $\epsilon$ and the purpose**
>
> Your question is insightful, and your intuition is exactly right. The parameter $\epsilon$ is indeed a standard damping term, introduced in Algorithm 1 primarily to ensure
> strict positive definiteness and thus the invertibility of the preconditioner matrix $V_t$. This is a common practice in adaptive optimization methods to guarantee numerical stability.
>
> However, in our specific implementation, the matrix $V_t$ is constructed using an exponential moving average (EMA), which provides inherent smoothing and accumulation of second-order information. As a result, the matrix $V_t$ becomes progressively better-conditioned during training, making the risk of singularity negligible after the initial iterations.
>
> After careful tuning, we found that choosing the damping parameter as small as possible always gave the best performance. Due to the stability provided by employing an EMA, we found we could set $\epsilon=0$ in our experiments without encountering any numerical issues. We compute the inverse square root $V_t^{-1 / 2}$ directly and robustly using Singular Value Decomposition (SVD). Setting $\epsilon = 0$ removes a tunable hyperparameter and allows the preconditioner to adapt more precisely to the loss geometry without any artificial damping.
>
> **Major question 2: Why is $\Lambda_t$ placed on the left side in Algorithm 1?**
>
> We place $\Lambda_t$ on the left side in Algorithm 1 to make the algorithm easier to understand, since this form is closer to the standard AdaGrad algorithm, where the parameters are treated as vectors and a diagonal preconditioner is used. Also, we point out that no matter on which side $\Lambda_t$ is placed, our theoretical analysis still holds. This is because, based on the algorithmic design, optimizing $f: \mathbb{R}^{m\times n} \to \mathbb{R}$, with $\Lambda_t$ on the right side is equivalent to optimizing $g: \mathbb{R}^{n\times m} \to \mathbb{R}$, where $g(W^\top) \equiv f(W)$, with $\Lambda_t$ on the left side. We thank the reviewer for the question and will add more explanation on this point in our revision.
>
> **Major question 3: Nonconvex analysis of ASGO**
>
> The answer to your insightful question is "Yes". By the quoted sentence, we mean that we expect to obtain a nonconvex convergence result for ASGO that is comparable to the Muon convergence rate that we established, since ASGO and Muon share some important algorithmic similarities. We think this would be an interesting topic for future research.
>
>
> **Major question 4: Wall-clock time analysis**
>
> We strongly agree that a fair comparison requires a thorough analysis of runtime, and we appreciate the opportunity to elaborate on the complexity of ASGO and provide detailed wall-clock time (WCT) measurements.
>
> To compare the WCT for ASGO with that for other algorithms, we conducted a new pre-training experiment for the 125M parameter GPT-2 model on the OpenWebText dataset with a significantly larger token budget. We trained the model for 1200 steps using a batch size of 64, 4 A6000 GPUs, a sequence length
> of 512, and 8 gradient accumulation steps. This setup corresponds to a total token budget of: $512 \times 64 \times 4 \times 8 \times 1200 \approx \mathbf{1 . 2 6}$ Billion tokens. This budget represents approximately 10 tokens per parameter, which attains the limit of our computational resources.
>
> As we mentioned regarding question 1, our initial version of ASGO uses SVD to compute the matrix inverse square root. While mathematically exact, this approach carries a significant computational burden. For the new experiments presented in this rebuttal, we have adopted the more efficient Newton-Schultz (NS) iterative method, which substantially reduces the wall-clock time of the optimizer step and makes the algorithm more practical. The following runtime analysis is based on this improved NS-based implementation.
>
> As shown in Thm 5.2 [6], computing $A^{-1 / 2}$ can be achieved by computing the sign function of a block matrix
>     $$
>     \operatorname{sign}\left(\left[\begin{array}{cc}
>     0 & A \\\\
>     I & 0
>     \end{array}\right]\right)=\left[\begin{array}{cc}
>     0 & A^{1 / 2} \\\\
>     A^{-1 / 2} & 0
>     \end{array}\right].
>     $$
>     This implies that computing the inverse square root in ASGO shares the same order of complexity as computing the matrix sign function in Muon. However, since the NS iteration in our case is applied to a matrix twice the dimension, the per-iteration cost is approximately double that of Muon.
>
> **Empirical Wall-Clock Time Analysis.** To provide a concrete analysis, we measured the WCT under the exact settings of our 1.26B token experiment.
> | **Optimizer** | **Single Train Step WCT** | **Time Difference** |
> | :--- | :---: | :---: |
> | AdamW | 5.451 sec | 0 sec |
> | Muon | 5.519 sec | 0.068 sec |
> | **ASGO (Ours)** | 5.947 sec | 0.468 sec |
>
> Table 1: Wall-clock time per training step, difference compared with AdamW update.
>
> - Single-Step Optimizer Cost: For a fair comparison of convergence, we use 10 iterations of the NS algorithm for ASGO instead of 5 iterations used for Muon's update. The measured WCT for a single train step (including forward, backward, gradient accumulation passes, and optimizer step) is shown in Table 1.
>
> Although computing the inverse square root of $V_t$, in ASGO's optimizer step is more expensive than in AdamW, the forward/backward pass is the dominant bottleneck. In our setup, a full training step in the ASGO update only costs 0.468s more than the AdamW update. This is acceptable in the training process. As a result, the total end-to-end training time is only marginally affected. Table 2 shows the total time to complete the 1200-step training run.
>
> | **Metric** | **AdamW** | **Muon** | **ASGO (Ours)** |
> | :--- | :---: | :---: | :---: |
> | WCT | 127.76 min | 129.01 min | 133.50 min |
> | Train Loss | 3.73 | 3.58 | 3.53 |
>
> Table 2: The total Wall-clock Time (WCT) and final training loss for training 1200 steps.
>
> These tables show that although AdamW is computationally cheapest at the optimizer level, the total training times for all three methods are very similar. This demonstrates that the total training time may not be a main concern for ASGO.
>     Note that ASGO achieves the smallest training loss while AdamW achieves the largest loss.
>
>
>
> **Minor Suggestions**
>
> We thank the reviewer for carefully reading our submission and suggesting ways to improve its clarity, which we will follow.
>
> **References**
>
> [1] John Duchi, Elad Hazan, and Yoram Singer. Adaptive subgradient methods for online learning and stochastic optimization. Journal of machine learning research 12.7 (2011).
>
> [2] Elad Hazan, Amit Agarwal, and Satyen Kale. Logarithmic regret algorithms for online convex optimization. Machine Learning 69.2 (2007): 169-192.
>
> [3] Vineet Gupta, Tomer Koren, and Yoram Singer. Shampoo: Preconditioned stochastic tensor optimization. International Conference on Machine Learning. PMLR, 2018.
>
> [4] Amit Attia and Tomer Koren. Sgd with adagrad stepsizes: Full adaptivity with high probability to
> unknown parameters, unbounded gradients and affine variance. International Conference on Machine Learning. PMLR, 2023.
>
> [5] Keller Jordan, Yuchen Jin, Vlado Boza, Jiacheng You, Franz Cesista, Laker Newhouse, Jeremy Bernstein. Muon: An optimizer for hidden layers in neural networks
>
> [6] Nicholas J. Higham. Functions of Matrices: Theory and Computation.

---

> > ### Author Response · Authors · 2025-08-05
> > **Follow-up Results**
> >
> > During this discussion period, we wanted to provide new, large-scale experimental results to further convince the reviewer of the effectiveness and scalability of our proposed method, ASGO.
> >
> > These results are from the large-scale 1.26B token GPT-2 pre-training experiment that we detailed in our response regarding the wall-clock time analysis. To ensure a fair comparison of final performance in that run, we performed a new Hyperparameter Optimization (HPO) for all three optimizers:
> >
> >
> > AdamW: lr in $\\{5 e^{-4},1 e^{-3},5 e^{-3},1 e^{-2}\\}$
> >
> > Muon: lr in $\\{1 e^{-3},5 e^{-3},1 e^{-2},5 e^{-2}, 1 e^{-1}\\}$
> >
> > ASGO: lr in $\\{1 e^{-3},5 e^{-3},1 e^{-2},5 e^{-2}, 1 e^{-1}\\}$
> >
> > | Optimizer | **315M** | **630M** | **945M** | **1.26B** |
> > | :--- | :---: | :---: | :---: | :---: |
> > | AdamW | 4.71 | 4.05 | 3.79 | 3.73 |
> > | Muon | 4.17 | 3.77 | 3.62 | 3.58 |
> > | **ASGO** | 4.11 | 3.74 | 3.58 | 3.53 |
> >
> > Table 1: Training loss at different checkpoints for the 125M GPT-2 model on OpenWebText.
> >
> > As Table 1 clearly indicates, the structure-aware preconditioning methods (Muon and our ASGO) consistently achieve a lower final training loss compared to the vector-based AdamW optimizer when trained on a larger token budget. This new experiment provides strong empirical evidence for our paper's central thesis: that leveraging the matrix structure of parameters is beneficial for training DNNs.
> >
> > We hope these new results help to address any remaining concerns about the practical viability and benefits of ASGO. We sincerely welcome any further questions and look forward to the discussion.

---

> > ### Comment · Reviewer_Vxtn · 2025-08-05
> >
> > I would like to thank the authors for their detailed reply and experiments. However, the current theoretical results are not convincing enough for me. My major concerns (two key weaknesses) remain. It is unclear whether providing an analysis for this algorithm with projection is straightforward, and whether the claimed theoretical 'benefits' over SGD would still hold under such conditions.

---

> > > ### Author Response · Authors · 2025-08-06
> > > **Reply to Reviewer Vxtn (R2, Part 1/2)**
> > >
> > > Thanks for the reply. We are happy to further clarify how we intend to revise our paper to address
> > > the reviewer's major concerns (i.e., two key weaknesses).
> > >
> > > **1. Regarding the Projection**
> > >
> > > We note that incorporating projection into the kind of convex analysis that we propose to add to our paper is an established technique that has been employed, for example, in the proof of Proposition 3 in [1] and Theorem 2 in [2]. We adopt their idea here. To formalize how to employ this technique for ASGO, we consider that we search for a solution in a closed convex set $\mathcal{W} \in \mathbb{R}^{m\times n}$ with at least one optimum $W_{\*}$. We can incorporate the projection operation by first replacing the algorithm update (line 8 in Algorithm 1) with
> > > $$
> > > Z_{t+1} = W_t - \eta_t \Lambda_t^{-1} G_t  \quad \text{and} \quad
> > > W_{t+1} = \Pi_{\Lambda_t}^{\mathcal{W}}\left( Z_{t+1} \right),
> > > $$
> > > where we follow the notation in [2] and use the projection operator $ \Pi_{\Lambda_t}^{\mathcal{W}}: \mathbb{R}^{m\times n} \to \mathbb{R}^{m\times n} $, defined as
> > >
> > > $$
> > > \Pi_{\Lambda_t}^{\mathcal{W}}(X) = \arg \min_{Y \in \mathcal{W}} || X - Y || _{\Lambda_t}^2 ,
> > > $$
> > >
> > > where $ || \cdot ||_{\Lambda_t} $ is defined in our Section 3.1.
> > >
> > > Note that this is almost the same as the standard Euclidean projection, only changing the distance metric from Frobenius norm to $ || \cdot ||_{\Lambda_t} $.
> > >
> > > Since we assume that there exists at least one optimum $ W_{\*} $ inside $ \mathcal{W} $ and $ W_{t+1} $ is the projection of $ Z_{t+1} $ onto the convex set $ \mathcal{W} $, based on the non-expansiveness of the projection operator, it holds that
> > > $$
> > > \mathrm{tr}\left( \Delta W_{t+1} \Lambda_t \Delta W_{t+1} \right) \leq \mathrm{tr}\left( \Delta Z_{t+1} \Lambda_t \Delta Z_{t+1} \right) ,
> > > $$
> > > where $ \Delta W_{t+1} = W_{t+1} - W_{\*} $ and $ \Delta Z_{t+1} = Z_{t+1} - W_{\*} $. With this inequality, the current proof for the theorems holds directly because we can rewrite the first inequality in Equation (6) in the paper (line 828 in Appendix E in the supplementary file) by
> > > $$
> > > 2\sum_{t=0}^{T-1} \mathbb{E}[f(W_t)] -f(W_*)
> > >     \le \frac{1}{\eta} \mathbb{E} \left[ \sum_{t=0}^{T-1} \mathrm{tr}\left( \Delta W_{t}^\top \Lambda_t \Delta W_{t} - \Delta Z_{t+1}^\top \Lambda_t \Delta Z_{t+1} \right) \right] + \eta \mathbb{E}\left[ \sum_{t=0}^{T-1} \mathrm{tr}\left( G_t^\top \Lambda_t^{-1} G_t \right) \right] .
> > > $$
> > > Then, using the non-expansiveness, we have
> > > $$
> > > 2\sum_{t=0}^{T-1} \mathbb{E}[f(W_t)] -f(W_*) \le \frac{1}{\eta} \mathbb{E} \left[ \sum_{t=0}^{T-1} \mathrm{tr}\left( \Delta W_{t}^\top \Lambda_t \Delta W_{t} - \Delta W_{t+1}^\top \Lambda_t \Delta W_{t+1} \right) \right] + \eta \mathbb{E}\left[ \sum_{t=0}^{T-1} \mathrm{tr}\left( G_t^\top \Lambda_t^{-1} G_t \right) \right] ,
> > > $$
> > > which is exactly the same as the original inequality, and thus, all the other analyses still hold. **With this projection onto a bounded convex set, we have that $ D_{\mathrm{op}} $ and $ D_{\mathrm{F}} $ are constants, and all the convergence rates hold the same, which should address the reviewer's concern.**
> > >
> > > We will include this discussion on the projection operation and the proof described above in the final version of the paper to strengthen the theoretical soundness.

---

> > > > ### Author Response · Authors · 2025-08-06
> > > > **Reply to Reviewer Vxtn (R2, Part 2/2)**
> > > >
> > > > **2. Regarding the Comparison with SGD**
> > > >
> > > > - We agree with the reviewer that the convergence of SGD has been established under less restrictive settings. However, we would like to first respectfully note that, as we discussed in the previous rebuttal, we didn't claim that the comparison between SGD and ASGO in the paper is a rigorous comparison, nor did we claim that ASGO must have benefits compared to SGD.
> > > > As we noted in Line 202 of the paper: "we further compare the $ \mathcal{O}(1/T) $ term here to provide some more intuition...", the comparison with SGD in our paper provides intuition for why we believe that ASGO can utilize the structured properties (e.g., low-rank gradients, since the upper bound of ASGO is relatively smaller when the property is satisfied), but **not a formal claim that ASGO is strictly better than SGD**.
> > > > - For a similar kind of comparison with SGD that provides intuition, we kindly refer the reviewer to [1,4].
> > > > In particular, in [1], the authors also compare the convergence upper bounds of AdaGrad (which requires a bounded domain for a rigorous proof, as we do) and SGD in a specific circumstance under
> > > > convex nonsmooth settings to give some intuition as to why and when AdaGrad can perform well. These comparisons are not fully rigorous as well, but provide intuition for when and why the algorithm may perform well.
> > > >
> > > > We hope these clarifications can address the reviewer's concerns about the theoretical properties.
> > > >
> > > > **References**
> > > >
> > > > [1] John Duchi, Elad Hazan, and Yoram Singer. Adaptive subgradient methods for online learning and stochastic optimization. Journal of machine learning research 12.7 (2011).
> > > >
> > > > [2] Elad Hazan, Amit Agarwal, and Satyen Kale. Logarithmic regret algorithms for online convex optimization. Machine Learning 69.2 (2007): 169-192.
> > > >
> > > > [3] Vineet Gupta, Tomer Koren, and Yoram Singer. Shampoo: Preconditioned stochastic tensor optimization. International Conference on Machine Learning. PMLR, 2018.
> > > >
> > > > [4] Jeremy Bernstein, et al. SignSGD: Compressed Optimisation for Non-Convex Problems. International conference on machine learning. PMLR, 2018.

---

> > > > > ### Comment · Reviewer_Vxtn · 2025-08-06
> > > > >
> > > > > Thank you to the authors for their response regarding the use of projection, which makes the results in Section 4 more sound. I am curious whether the projection could be incorporated into the smooth setting under the current definition of smoothness.

---

> > > > > > ### Author Response · Authors · 2025-08-07
> > > > > > **Reply to Reviewer Vxtn (R3)**
> > > > > >
> > > > > > Thanks for the follow-up question. The answer is **yes**, the projection can be incorporated directly into the smooth setting, and our theoretical framework is structured to support this seamlessly.
> > > > > >
> > > > > > The core reason is that our proof for the smooth case (Theorem 3) is built directly upon the convergence result of the non-smooth case (Theorem 1). The transition between the two settings relies only on a few key technical lemmas that are entirely independent of the projection operator.
> > > > > >
> > > > > > To be more specific:
> > > > > >
> > > > > > 1. The proof starts with Lemma 7 (Appendix F), our key technical bridge. This lemma divides the average expected function suboptimality, $\frac{1}{T} \sum_{t=0}^{T-1} \mathbb{E}\left[f\left(W_t\right)-f\left(W_*\right)\right]$, into two major components: (i) a gradient norm term, and (ii) a variance term. Note that we omit the $ \epsilon $ term in the rebuttal discussion since it is usually very small.
> > > > > >
> > > > > > 2. To achieve the final bound in Theorem 3, we bound each of these terms. The lemmas used for this step - the matrix inequality in Lemma 8 and the variance bound in Lemma 9 - are
> > > > > > based only
> > > > > > on algebra and probability facts. In the proof of Theorem 3, we further bound the gradient norm term by utilizing the smoothness property $ || \nabla f(W) ||_{L^{-1}}^2 \leq 2(f(W) - f^*) $. All of these operations are not affected by the projection.
> > > > > >
> > > > > >
> > > > > > Therefore, once we establish Theorem 1 in the projected setting (as discussed in our previous response), Theorem 3 naturally applies, and hence, the convergence result for the smooth case still holds.
> > > > > >
> > > > > >
> > > > > > Thank you very much for raising the issue of incorporating the projection into our analysis of the smooth case, i.e., Theorem 3. We will include this extension in a revised version of our paper and acknowledge that it was motivated by a question from an anonymous reviewer. We also hope that our response fully satisfies your curiosity.

---

> > > > > > > ### Comment · Reviewer_Vxtn · 2025-08-08
> > > > > > >
> > > > > > > Thank you to the authors for their response regarding the projections. I have already raised my score. Overall, the paper demonstrates strong merits in its experiments and the novel perspective it offers, although I believe the current comparison with SGD remains somewhat weak.

---

> > > > > > > > ### Author Response · Authors · 2025-08-09
> > > > > > > > **Reply to Reviewer Vxtn, Thanks!**
> > > > > > > >
> > > > > > > > We sincerely appreciate your constructive responses and recognition of our work. We fully understand your expectation of a rigorous comparison between ASGO and SGD. For now, we only consider it an intuitive explanation for when and how ASGO can perform well, but it will definitely be an interesting future direction to demonstrate a more rigorous comparison.
> > > > > > > > Thanks again for the help in improving the quality of the paper!

---

### Comment · Area_Chair_EL9w · 2025-08-06

Thanks for the 3 reviewers that engaged in the discussions!

Dear reviewer hvYf,

  Can you look at the author rebuttal and other reviews and engage in the mandatory discussion of this paper please?

AC

---

### Author Response · Authors · 2025-08-09
**To all Reviewer, Thanks for all the meaningful responses and suggestions!**

Dear Reviewers,

We sincerely appreciate the time and effort you have invested in reviewing our manuscript and providing such insightful suggestions and comments. We are also thrilled to know that reviewers acknowledged that our paper:

- Provides solid convergence theorems for the proposed algorithm ASGO (Reviewer Vxtn, WuZT, hvYf)
- Explains the potential benefits of ASGO through discussions (Reviewer Vxtn, WuZT, hvYf)
- Demonstrates promising empirical performance (Reviewer Vxtn, WuZT, hvYf)

During the rebuttal period, we have diligently addressed each of the reviewers' concerns in detail by introducing the following modifications:

- On the theory side, we provide a more detailed analysis with the projection operation incorporated, making the convergence theorems stronger and more convincing.

- On the empirical side, we provide experiments on pretraining a 125M GPT-2 model on 1.26B tokens, demonstrating the practical performance of ASGO on a larger scale. Moreover, we conduct a runtime analysis of ASGO to show that the additional computation burden of ASGO is acceptable (only around 5\% more total training time in our setting). We believe the two additional experiments should address the reviewer's concerns on the empirical validation part of our paper.


Thanks again for your dedication. We sincerely appreciate all your constructive feedback and suggestions.


Sincerely,

Authors of submission 18201

---

### Note · Authors · 2025-08-13

Dear SAC, AC, and Reviewers,


We are very pleased that Reviewers (Vxtn, WuZT, hvYf) acknowledged that our paper:
(1) provides solid convergence theorems for the proposed algorithm ASGO,
(2) explains the potential benefits of ASGO through discussions,
 and
(3) demonstrates promising empirical performance.

During the rebuttal period, we have diligently addressed each of the reviewers' concerns in detail by introducing the following modifications:

* On the theory side, we provided a more detailed analysis that incorporates the projection operation for both smooth and non-smooth settings, making our theorems more robust. We also discussed interesting future directions that the reviewers mentioned.




* On the empirical side, we provided experiments on a larger-scale pretraining task, together with a detailed runtime analysis of ASGO to highlight the practical efficiency and performance of ASGO.

Additionally, we wish to summarize our discussion with Reviewer uDaJ, with whom we could not fully reach a consensus.

Initially, we successfully addressed foundational misunderstandings that Reviewer uDaJ raised about our characterization of adaptive gradient methods. The reviewer had no further theoretical questions after our clarification; hence, we believe that we have reached a consensus regarding the paper's theoretical contributions.



The remaining disagreements stem from:

- Experimental Scope: The reviewer insisted on broader experiments (e.g., AlgoPerf) and larger-scale experiments to prove the "generality" of our optimizer. We contend that our theoretical conclusion and current, popular LLM experiments are sufficient for demonstrating ASGO's practical value.

- Code Availability: The reviewer repeatedly criticized the lack of code. We have committed to open-source and reproducible code upon publication, which is standard practice.

Reviewer uDaJ's score of 2 is apparently based on these two points. While we respect the reviewer's opinion, we kindly but strongly disagree with this assessment, as we believe these demands fall outside the typical scope of evaluation for a paper of this nature.


We sincerely appreciate the time and effort you have invested in our submission and have full confidence in having a fair and rigorous decision process.


Sincerely,

Authors of submission 18201

---

### Decision · Program_Chairs · 2025-09-17

**Decision:**

Accept (poster)

**Comment:**

This paper proposes ASGO, a new adaptive approach for stochastic gradient optimization. ASGO differs from prior work Shampoo and K-FAC in that it uses a single matrix conditioner to condition the gradient updates of matrix-valued parameters. The authors provided a convergence theory of this in convex problems, and shown that the convergence bounds of this approach might be even better than Shampoo-style preconditioning and comparable with Muon but with better theoretical guarantees.

Reviewers raised the questions about empirical timing and the authors provided a version in the rebuttal that utilizes Newton-Schultz iterations to compute the matrix inverse square-root. After rebuttal, 3 reviewers are on the positive side, with ratings of 1 Accept and 2 borderline accepts. One reviewer still stands on the negative side. The main remaining issues are that the algorithm is only tested on a single LLM benchmark, and that the code was not available.

The AC believes that the availability of the code should not be used to hold against authors. And also that the paper's contribution is of theoretical nature hence can enjoy some leniency on the results. Thus, AC recommends acceptance of this paper.